# Key role of $e_g^*$ band broadening in nickel-based oxyhydroxides on coupled oxygen evolution mechanism

Haoyin Zhong [1], Qi Zhang [1], Junchen Yu [1], Xin Zhang[1], Chao Wu[2,3], Hang An[1], Yifan Ma[1], Hao Wang[1], Jun Zhang[1], Yong-Wei Zhang [4], Caozheng Diao [5], Zhi Gen Yu [4] ✉, Shibo Xi[2] ✉, Xiaopeng Wang [1,3] ✉ & Junmin Xue [1] ✉

A coupled oxygen evolution mechanism (COM) during oxygen evolution reaction (OER) has been reported in nickel oxyhydroxides (NiOOH)-based materials by realizing $e_g^*$ band (3d electron states with $e_g$ symmetry) broadening and light irradiation. However, the link between the $e_g^*$ band broadening extent and COM-based OER activities remains unclear. Here, $Ni_{1-x}Fe_xOOH$ (x = 0, 0.05, 0,2) are prepared to investigate the underlying mechanism governing COM-based activities. It is revealed that in low potential region, realizing stronger $e_g^*$ band broadening could facilitate the $^*OH$ deprotonation. Meanwhile, in high potential region where the photon utilization is the rate-determining step, a stronger $e_g^*$ band broadening would widen the non-overlapping region between $d_z^2$ and $a_{1g}^*$ orbitals, thereby enhancing photon utilization efficiency. Consequently, a stronger $e_g^*$ band broadening could effectuate more efficient OER activities. Moreover, we demonstrate the universality of this concept by extending it to reconstruction-derived X-NiOOH (X = $NiS_2$, $NiSe_2$, $Ni_4P_5$) with varying extent of $e_g^*$ band broadening. Such an understanding of the COM would provide valuable guidance for the future development of highly efficient OER electrocatalysts.

Electrochemical water splitting, encompassing the hydrogen evolution reaction (HER) and the oxygen evolution reaction (OER), constitutes a pivotal technology for addressing the intermittency issues related to renewable energy sources such as solar, wind and tidal power[1–4]. Among the two reactions, OER is regarded as the bottleneck due to its slow kinetics associated with multiple electron transfer steps and represents a key determinant of the overall energy conversion efficiency into chemical fuels. Thus, the development of high-performance OER electrocatalysts is of paramount importance to advance the field of sustainable energy conversion[5–8]. Currently, there are two widely recognized

mechanisms for the OER process, i.e., the adsorbate evolution mechanism (AEM), a metal redox reaction, and the lattice oxygen oxidation mechanism (LOM), which involves oxygen redox chemistry[9–12]. The AEM pathway typically involves the rate-determining step (RDS) of $^*OOH$ formation, whereas in the LOM pathway, the deprotonation step serves as the RDS[13–15]. The existence of the RDS poses a fundamental challenge to electron transfer efficiency during OER, impeding catalytic performance enhancement. As a result, substantial research efforts have been dedicated to optimizing OER reaction pathways in pursuit of highly efficient electrocatalysts[16–19].

[1]Department of Materials Science and Engineering, National University of Singapore, Singapore 117575, Singapore. [2]Institute of Sustainability for Chemical, Energy and Environment (ISCE2), Agency for Science, Technology and Research, Singapore 627833, Singapore. [3]College of Materials Science and Engineering, Sichuan University, Chengdu 610065, China. [4]Institute of High Performance Computing, Agency for Science, Technology and Research, Singapore 138632, Singapore. [5]Singapore Synchrotron Light Sources (SSLS), National University of Singapore, Singapore 117603, Singapore. ✉e-mail: yuzg@ihpc.a-star.edu.sg; xi_shibo@isce2.a-star.edu.sg; msewxia@nus.edu.sg; msexuejm@nus.edu.sg

Previously, the concept of a coupled oxygen evolution mechanism (COM) was introduced, which featured alternative metal/oxygen redox activities occurring throughout the oxygen evolution process[14]. The initiation of the COM mechanism was reliant upon both light irradiation and the broadening of the $e_g^*$ band in nickel oxyhydroxides (NiOOH)-based materials. In contrast to the traditional AEM pathway, the COM route involved direct O-O coupling at the oxygen states, bypassing the RDS step (i.e.,*OOH formation) in AEM. Also, the deprotonation process in the COM pathway involved proton transfer occurring at the metal bands, followed by light-induced electron transfer from the (M-O) orbitals to the $d_z^2$ orbital, resembling the proton transfer in AEM deprotonation. This potentially indicated a lower energy requirement for deprotonation in COM compared to LOM. As a result, the presented electron transfer mechanism proceeded through an optimized pathway, where deprotonation occurred at the metal bands and O-O coupling took place at the oxygen states. Therefore, the COM pathway offered promising avenues for developing OER electrocatalysts with more efficient catalytic performance. However, the relationship between the extent of $e_g^*$ band broadening and the resulting OER activities enhancement is not yet fully understood, particularly in terms of how the $e_g^*$ band broadening influences the RDS in the COM route. These significantly impedes researchers from further optimizing the OER activities via the COM pathway.

Herein, $Ni_{1-x}Fe_xOOH$ (x = 0, 0.05, 0,2) are synthesized as model materials to investigate the underlying mechanism governing the COM-based OER activities. It is revealed that Fe dopants could effectively tune the $NiO_6$ distortion in NiOOH, resulting in varying degrees of the $e_g^*$ band broadening. In the low potential region where *OH deprotonation serves as the RDS, increasing the extent of $e_g^*$ band broadening could significantly facilitate the *OH deprotonation. Meanwhile, in the high potential region where light absorption becomes the RDS, a stronger $e_g^*$ band broadening would lead to a wider non-overlapping region between $d_z^2$ and $a_{1g}^*$ orbitals. This greatly facilitates photon-induced electron transfer from (M-O) to empty $d_z^2$ orbital, leading to enhanced photon utilization efficiency. Consequently, the catalyst with higher extent of $e_g^*$ band broadening exhibits greater enhancement in OER activity under the COM route. Moreover, we observe a consistent relationship between the extent of $e_g^*$ band broadening and COM-based OER activity in the X-NiOOH (X = NiS_2, NiSe_2, Ni_5P_4) system, where strain effects modulate $e_g^*$ band

broadening, highlighting the universality of this concept across diverse materials. The insights gained from this study on COM could offer valuable guidance for the development of efficient OER electrocatalysts, thus promoting the advancement of energy conversion technologies.

## Results

### Identify COM pathway contributed OER activity promotion

It was revealed that 5% and 20% Fe could be successfully doped into the $Ni(OH)_2$ lattice[20]. More content of Fe dopants i.e., 40%, 60%, 100% would lead to the aggregation of Fe cations to form $Fe_2O_3$ (Supplementary Fig. 1). Since $Fe_2O_3$ is a well-studied photocatalyst, its presence might influence the electrochemical results based on the COM. Hence, to focus on investigating the impact of Fe dopants-induced $e_g^*$ band broadening on the COM, the $Ni_{1-x}Fe_xOOH$ (x = 0, 0.05, 0,2) are chosen as model samples. Fe-removed 1 M KOH electrolyte is employed to minimize the influence of Fe impurity on electrochemical measurements, (the experimental process for Fe removal is described in the Methods section)[21]. The pH value of the purified 1 M KOH is measured to be 13.65 and the error bars represent mean ± standard error (Supplementary Fig. 2). Figure 1a shows the linear sweep voltammetry (LSV) curves of $Ni_{1-x}Fe_xOOH$ (x = 0, 0.05, 0,2) under both light and dark conditions. Negative scan is conducted for the LSV measurement to avoid the influence of $Ni^{2+/3+}$ redox current on the OER activity. Hence the negative current peak due to the redox from $Ni^{3+}$ to $Ni^{2+}$ are observed for $Ni_{1-x}Fe_xOOH$ (x = 0, 0.05, 0,2) (Fig. 1a and Supplementary Fig. 3). Further detailed methodology for measuring the electrochemical activities is provided in the Methods section. Notably, $Ni_{0.8}Fe_{0.2}OOH$ exhibits the highest OER activity under dark condition, followed by $Ni_{0.95}Fe_{0.05}OOH$ with the second-highest activity and NiOOH with the lowest activity. For NiOOH under light irradiation, negligible current density variation is observed, which agrees well with our previous research[14]. Interestingly, a significant increase in current density can be observed for $Ni_{0.95}Fe_{0.05}OOH$ and $Ni_{0.8}Fe_{0.2}OOH$ under light condition. The overpotential drops for $Ni_{0.8}Fe_{0.2}OOH$ and $Ni_{0.95}Fe_{0.05}OOH$ at 10 mA cm$^{-2}$ between dark and light conditions are 22 mV and 11 mV, respectively (Fig.1b). To further investigate the contribution of COM to the enhancement of OER performance, the intrinsic activities of $Ni_{1-x}Fe_xOOH$ (x = 0.05, 0,2) are provided with the current density normalized to electrochemical surface area (ECSA,

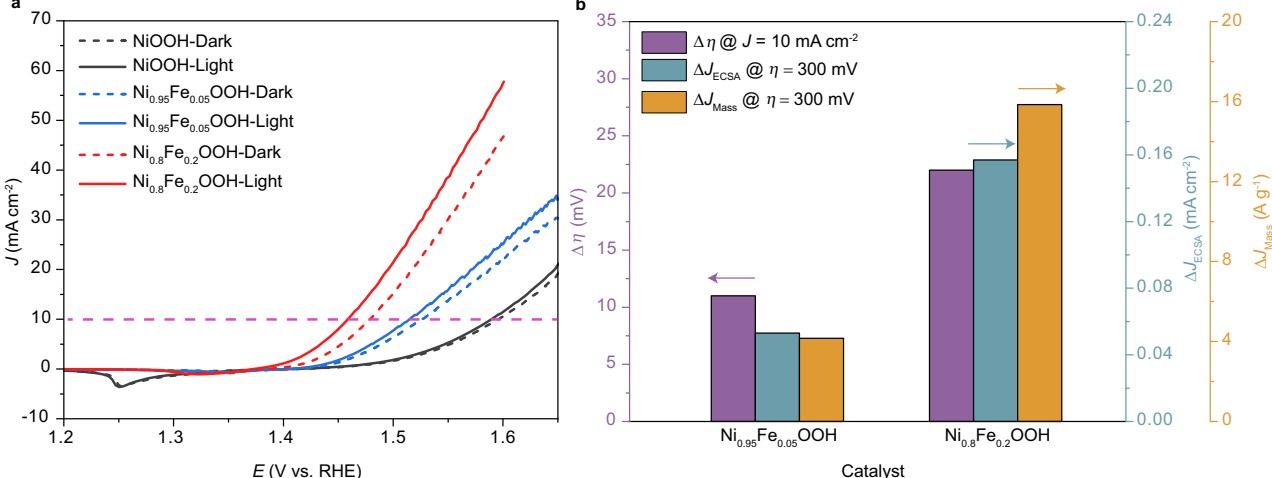

**Fig. 1 | Electrochemical characterization of $Ni_{1-x}Fe_xOOH$ (x = 0, 0.05, 0.2) for the OER under dark and light condition. a** LSV polarization curves of $Ni_{1-x}Fe_xOOH$ (x = 0, 0.05, 0.2) based on a backward scan conducted at a scan rate of 0.1 mV s$^{-1}$ under dark (dash line) and light (solid line) condition (without *iR*-correction). **b** Comparison of the overpotential drop ($\Delta\eta$) at a current density ($J$) of 10 mA cm$^{-2}$

(purple color), the current density improvement normalized to ECSA ($\Delta J_{ECSA}$) at an overpotential of 300 mV (cyan color), and the current density improvement normalized to loading mass ($\Delta J_{Mass}$) at an overpotential of 300 mV (orange color), respectively for $Ni_{1-x}Fe_xOOH$ (x = 0.05, 0.2).

Supplementary Fig. 4) and loading mass. As shown in Fig.1b, the ECSA-normalized current density difference between light and dark conditions is 0.157 mA cm$^{-2}$ for Ni$_{0.8}$Fe$_{0.2}$OOH and 0.053 mA cm$^{-2}$ for Ni$_{0.95}$Fe$_{0.05}$OOH at an overpotential of 300 mV. Meanwhile, the loading mass-normalized current density improvement is 15.846 A g$^{-1}$ for Ni$_{0.8}$Fe$_{0.2}$OOH and 4.159 A g$^{-1}$ for Ni$_{0.95}$Fe$_{0.05}$OOH, respectively. Based on these results, it can be concluded that the intrinsic OER activity increment of Ni$_{0.8}$Fe$_{0.2}$OOH under the COM route is greater than that of Ni$_{0.95}$Fe$_{0.05}$OOH.

In our previous investigation, we have established that the COM predominantly governs the electrochemical process in low potential region, where there is an ample supply of photons to initiate electron transfer from (M-O) to $d_z^2$, prompting geometric transition from NiO$_6$ octahedra to NiO$_4$ square planar structures[14]. In high potential region, the limited photon utilization would hinder further contribution of COM to the OER activity, thus serving as the new RDS (Fig. 2a). Therefore, it is of great importance to firstly investigate the two possible RDS in low potential region and high potential region, respectively. The deprotonation of ˙OH in the COM route closely parallels that in the AEM route, with the primary distinction being the absence of electron transfer during the deprotonation step in the COM pathway[14]. As such, the ˙OH deprotonation ability in the AEM route could be used to estimate that in the COM route. Therefore, pulse-voltammetry measurements are conducted to investigate the ˙OH deprotonation ability of Ni$_{1-x}$Fe$_x$OOH (x = 0.05, 0,2) within the low potential range (Supplementary Fig. 5, detailed protocol is described in the Methods section). As shown in Fig. 2b, the total stored charge $Q_{ECSA}$ varies linearly with the applied potential for both Ni$_{0.95}$Fe$_{0.05}$OOH and Ni$_{0.8}$Fe$_{0.2}$OOH. During OER process, the oxidative charge is accumulated on the surface of catalysts via the deprotonation step[22]. Hence, the slopes of these fitted lines are calculated to elucidate the rate of oxidative charge to the potential, thereby reflecting the ˙OH deprotonation ability[20]. The slope value for Ni$_{0.8}$Fe$_{0.2}$OOH is 24.257, which is significantly higher than the slope value of 7.450 for Ni$_{0.95}$Fe$_{0.05}$OOH. This discrepancy suggests that the ˙OH deprotonation ability of Ni$_{0.8}$Fe$_{0.2}$OOH in the COM route is much stronger than that of Ni$_{0.95}$Fe$_{0.05}$OOH.

Subsequently, we delve into the analysis of photon utilization efficiency. Within the high potential range, the photon utilization to trigger the conversion from NiO$_6$ octahedron to NiO$_4$ square planar would gradually become the RDS. In this work, a simulated solar source matching AM 1.5 G was used as the light source with a light irradiation intensity of 100 mW cm$^{-2}$. As the potential increases, the COM would become limited due to insufficient photon, leading to the co-existence of COM and AEM route during OER process. Hence, further increasing the applied potential, the promotion of OER activity would be mainly

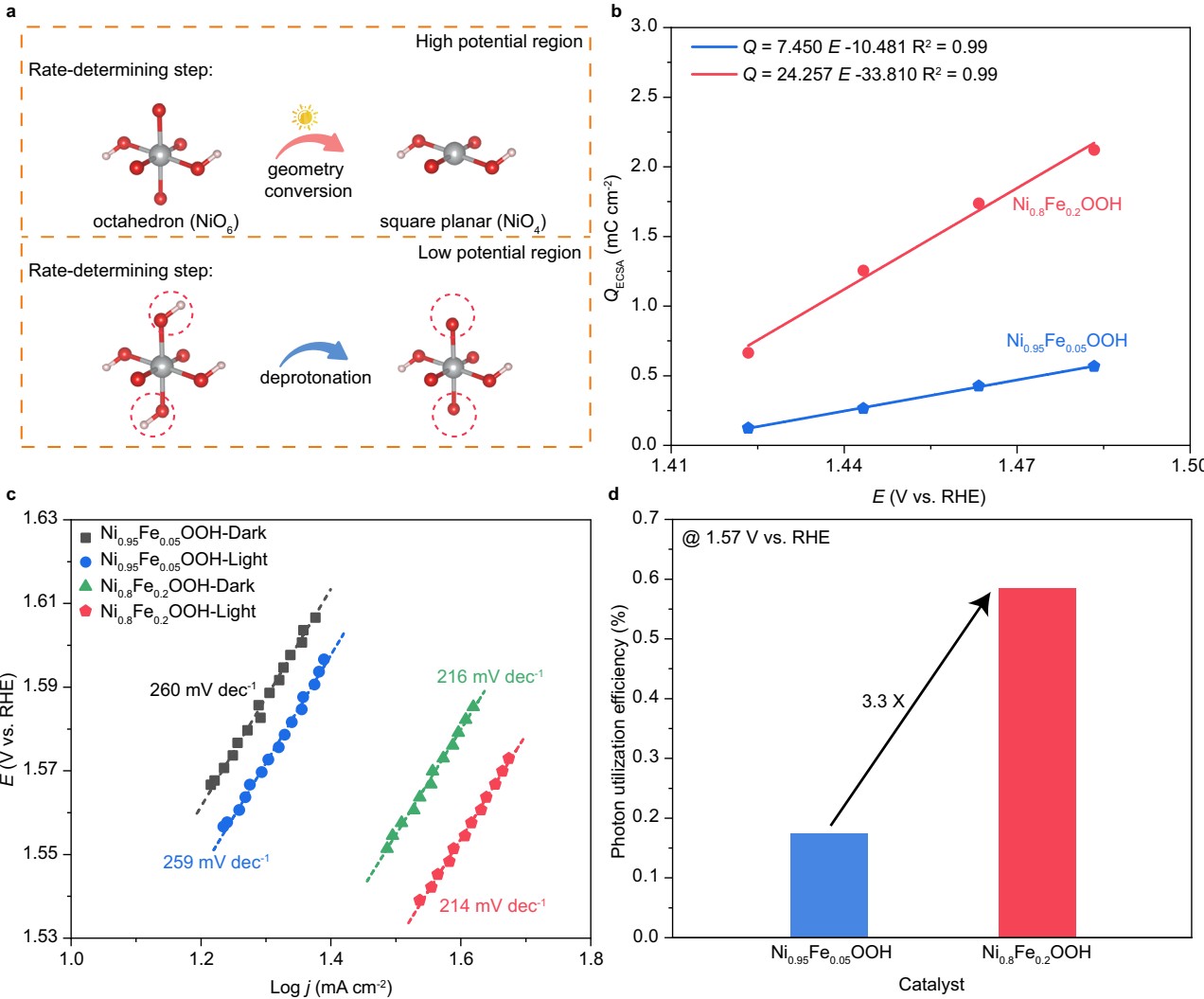

**Fig. 2 | Analysis of ˙OH deprotonation and photon utilization efficiency for Ni$_{1-x}$Fe$_x$OOH (x = 0.05, 0.2).** **a** Schematic illustration of the rate-determining step in low potential region and high potential region under COM route. **b** Charge versus potential for Ni$_{1-x}$Fe$_x$OOH (x = 0.05, 0.2) from pulse-voltammetry (without *iR*-correction). **c** Tafel slope for Ni$_{1-x}$Fe$_x$OOH (x = 0.05, 0.2) in high potential region. **d** Photon utilization efficiency for Ni$_{1-x}$Fe$_x$OOH (x = 0.05, 0.2) at 1.57 V vs. RHE.

contributed by AEM rather than COM. This could be understood from Tafel slope analysis for $Ni_{0.95}Fe_{0.05}OOH$ and $Ni_{0.8}Fe_{0.2}OOH$ in the high potential region, revealing identical slope values under both light and dark conditions (Fig. 2c). Based on the COM route, two photons would participate into the evolution of one oxygen molecule, concomitant with the transfer of four electrons to the external circuit. Therefore, we could estimate the number of participated photons based on the enhanced OER current between light and dark conditions in the high potential region. The photon utilization efficiency hence can be derived by normalizing the number of participated photons to the whole number of photons under 100 mW cm$^{-2}$ light irradiation (Detailed calculation information is provided in the Methods section and Supplementary Fig. 6). Here, the photon utilization efficiency for $Ni_{0.95}Fe_{0.05}OOH$ and $Ni_{0.8}Fe_{0.2}OOH$ is compared at the potential of 1.57 V vs. RHE. The calculated photon utilization efficiency is 0.584% for $Ni_{0.8}Fe_{0.2}OOH$, which is nearly 3.3 times than that of $Ni_{0.95}Fe_{0.05}OOH$, showing an efficiency of 0.175% (Fig. 2d).

### Different $e_g^*$ band broadening extent in Fe-doped NiOOH system

The extent of $e_g^*$ band broadening in $Ni_{1-x}Fe_xOOH$ (x = 0, 0.05, 0,2) is discussed utilizing a suite of density functional theory (DFT) calculations, Raman spectroscopy, and extended X-ray absorption fine structure (FT-EXAFS) results. Firstly, the DFT calculation is conducted using $Ni_{1-x}Fe_xOOH$ (x = 0, 0.05, 0,2) as computational models (Supplementary Fig. 7) to systematically investigate the effect of Fe dopants on tuning $NiO_6$ distortion and $e_g^*$ band broadening. Upon introducing Fe into NiOOH (extract from $Ni_{0.8}Fe_{0.2}OOH$ model, Supplementary Fig. 7), noticeable extensions of Ni-O bonds along the z-axis are observed, with values increasing from 1.871 Å to 2.034 Å and from 1.894 Å to 1.997 Å, respectively, as compared to NiOOH (Fig. 3a). Then, the $e_g^*$ band partial density of states (PDOS) of $Ni_{1-x}Fe_xOOH$ (x = 0, 0.05, 0,2) is calculated. As shown in the Fig. 3b, the $e_g^*$ band PDOS peak becomes broader with lower intensity from NiOOH, $Ni_{0.95}Fe_{0.05}OOH$, to $Ni_{0.8}Fe_{0.2}OOH$. This trend indicates a stronger $e_g^*$ band broadening associated with higher concentrations of Fe dopants. Raman spectroscopy of $Ni_{1-x}Fe_xOOH$ (x = 0, 0.05, 0,2) shows two typical peaks at 472 cm$^{-1}$ and 555 cm$^{-1}$ (Fig. 3c). Notably, the intensity (I) ratio between these two peaks can be used to identify the crystal structure of NiOOH[23]. The measured $I_{472}/I_{550}$ value here is 2.81 for NiOOH, 2.07 for $Ni_{0.95}Fe_{0.05}OOH$, and 1.40 for $Ni_{0.8}Fe_{0.2}OOH$, respectively, indicating a significantly increased level of disorder in the crystal structure of NiOOH with higher Fe dopants concentration. Furthermore, FT-EXAFS analysis demonstrates a reduction in the intensity of the Ni-O bond peak from $Ni(OH)_2$, $Ni_{0.95}Fe_{0.05}(OH)_2$, to $Ni_{0.8}Fe_{0.2}(OH)_2$ (Fig. 3d). The lower Ni-O bond peak intensity is corresponded to the increased extent of $NiO_6$ octahedron distortion. Neither the peak intensity around 6 Å nor the Ni-Ni bond peak intensity show great difference, indicating that there is no significant particle size difference among these samples[20]. Hence, the structural difference among these $Ni_{1-x}Fe_x(OH)_2$ (x = 0, 0.05, 0.2) should be ascribed to the doping effect instead of strain effect. Moreover, the energy difference between $d_{x^2-y^2}$ and $d_{z^2}$, calculated based on the band center difference between $d_{x^2-y^2}$ and $d_{z^2}$, is 0.21 eV, 0.22 eV, and 0.25 eV for NiOOH, $Ni_{0.95}Fe_{0.05}OOH$, and $Ni_{0.8}Fe_{0.2}OOH$, respectively (Supplementary Figs. 8–10). This provides strong evidence for the greater $e_g^*$ band broadening with higher concentration of Fe dopants, which agrees well with our prior reported X-ray absorption spectroscopy (XAS) analysis (Fig. 3e)[20]. Hence, both experimental and theoretical results demonstrate that the introduction of Fe dopants into NiOOH induces varying degrees of $NiO_6$ octahedron distortion, resulting in distinct levels of $e_g^*$ band broadening around the Fermi level, with the order being $Ni_{0.8}Fe_{0.2}OOH > Ni_{0.95}Fe_{0.05}OOH > NiOOH$.

### Effects of $e_g^*$ band broadening on COM

As demonstrated, the main structural distinction among $Ni_{1-x}Fe_xOOH$ (x = 0.05, 0.2) lies in the degree of $NiO_6$ octahedron distortion, leading

to varying degrees of $e_g^*$ band broadening. Moreover, both Ni K-edge and L-edge XAS measurements show that there is negligible structural difference for the samples after OER under dark or light conditions, confirming the reversibility of the electrode surface during the COM-based OER process (Supplementary Fig. 11). In this section, the effect of $e_g^*$ band broadening on the ·OH deprotonation in $Ni_{1-x}Fe_xOOH$ (x = 0.05, 0.2) is investigated. Here, the slope obtained from pulse-voltammetry measurement serves as a representative metric for ·OH deprotonation ability, while the white line intensity extracted from Ni K-edge spectra is employed to indicate the degree of $e_g^*$ band broadening[20]. As shown in Fig. 4a, the $Ni_{0.8}Fe_{0.2}OOH$ with stronger $e_g^*$ band broadening shows enhanced ·OH deprotonation ability compared to the $Ni_{0.95}Fe_{0.05}OOH$. Meanwhile, the calculated reaction free energies of the ·OH deprotonation process for $Ni_{0.8}Fe_{0.2}OOH$ following the COM route is also lower than that of the $Ni_{0.95}Fe_{0.05}OOH$, which is consistent with the pulse-voltammetry results (Supplementary Fig. 12). The enhanced proton transfer ability ultimately results in a substantial improvement in the OER activity under light irradiation within the low potential region, where the ·OH deprotonation serves as the RDS (Fig. 4b).

With the applied potential increased, the COM-based OER activity would be limited by the photon utilization efficiency. Our prior research demonstrated that the non-overlapping region between $d_{z^2}$ and $a_{1g}^*$ orbitals played a key role in the photon absorption process, which was greatly related to the $e_g^*$ band broadening[14]. To unveil the underlying science of the extent of $e_g^*$ band broadening and enhanced OER activities in the high potential region, the relationship between the $e_g^*$ band broadening and the photon utilization efficiency is investigated. In theory, if the non-overlapping region between $d_{z^2}$ and $a_{1g}^*$ orbitals is broader, it would make the electron transfer from (M-O) to $d_{z^2}$ easier under light irradiation. This, in turn, would result in a higher photon utilization efficiency. As shown in Fig. 4c, the $d_{z^2}$ orbital for NiOOH is completely overlapped with the $a_{1g}^*$ orbital. Hence, electrons would be unfeasible to transfer from (M-O) to $d_{z^2}$. After doping Fe into NiOOH, a distinctive non-overlapping region emerges between $d_{z^2}$ and $a_{1g}^*$ for both $Ni_{0.95}Fe_{0.05}OOH$ and $Ni_{0.8}Fe_{0.2}OOH$, indicating the possible electron transfer under light irradiation. More importantly, the broadness of the non-overlapping region increases from 0.65 eV for $Ni_{0.95}Fe_{0.05}OOH$ to 0.84 eV for $Ni_{0.8}Fe_{0.2}OOH$. The broadening of the non-overlapping region could be ascribed to the stronger extent of $e_g^*$ band broadening induced by Fe dopants (Fig. 4d). Subsequently, the relationship between the broadness of nonoverlapping region and the corresponding photon utilization efficiency is analyzed. As shown in Fig. 4e, with the increase of broadness of the non-overlapping region, the photon utilization efficiency is significantly promoted from 0.175% to 0.584%. To further confirm such effect, LSV measurements of $Ni_{0.95}Fe_{0.05}OOH$ and $Ni_{0.8}Fe_{0.2}OOH$ under light with wavelengths of 575 nm, 475 nm, and 365 nm are performed. The $Ni_{0.8}Fe_{0.2}OOH$ with stronger $e_g^*$ band broadening exhibits higher current density promotion at 1.57 V vs. RHE between dark and light conditions at each wavelength compared to $Ni_{0.95}Fe_{0.05}OOH$ (Supplementary Fig. 13). From a calculated perspective, the reaction free energy of the light-dominated step for $Ni_{0.8}Fe_{0.2}OOH$ is 0.89 eV, much lower than that of the $Ni_{0.95}Fe_{0.05}OOH$ (0.94 eV) (Supplementary Fig. 12). As such, these results prove that increasing the non-overlapping region via tuning $e_g^*$ band broadening would facilitate the electron transfer from (M-O) to $d_{z^2}$ under light irradiation, leading to higher photon utilization efficiency (Fig. 4f).

### Increasing $e_g^*$ band broadening can promote participation of COM – a universal concept

Based on the above discussion, it is anticipated that increasing the extent of $e_g^*$ band broadening in NiOOH-based materials could promote the participation of COM during OER, realizing higher OER activities. Our previous work revealed that the $e_g^*$ band broadening

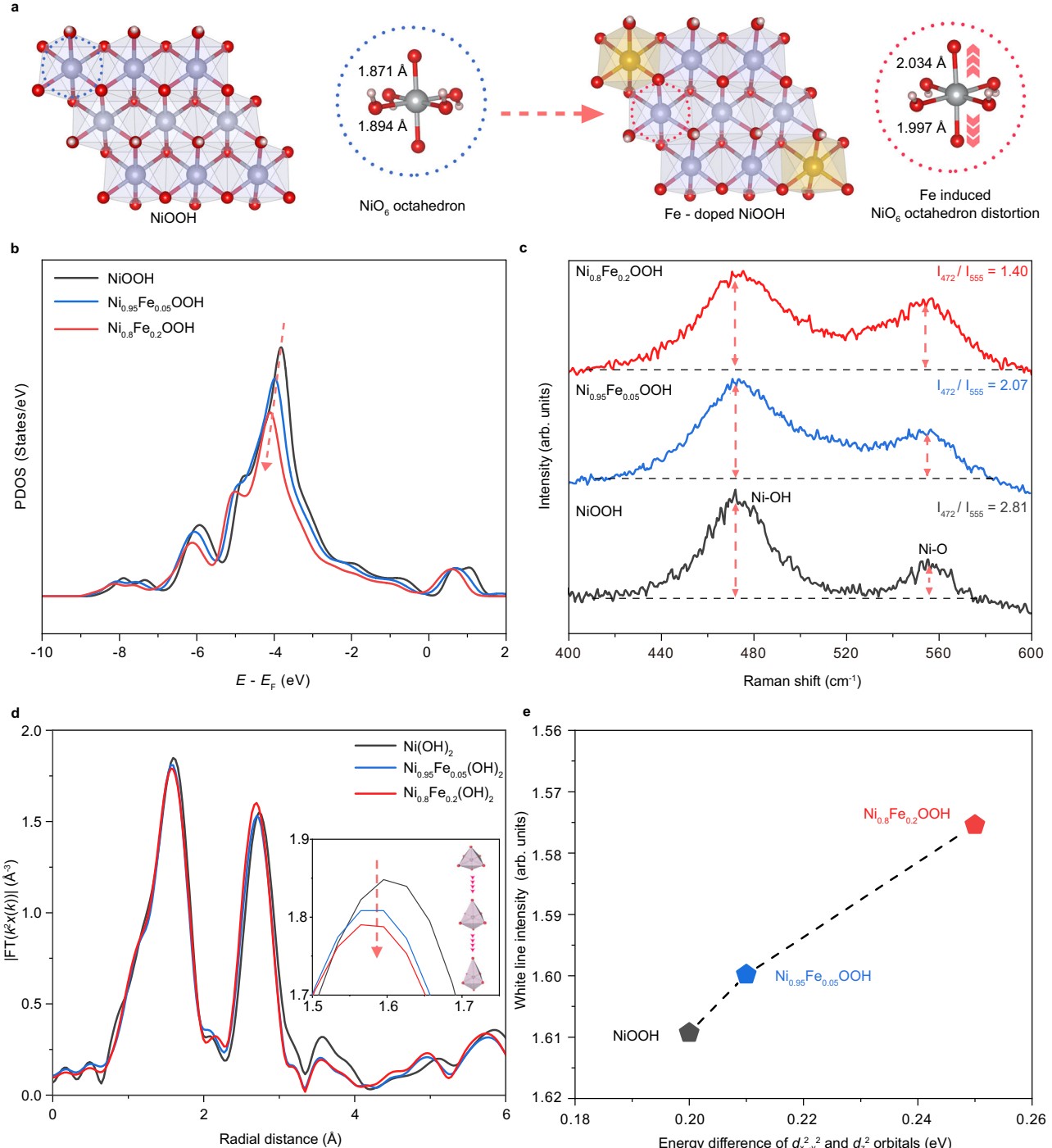

**Fig. 3 | Analysis of $e_g^*$ band broadening extent of Ni$_{1-x}$Fe$_x$OOH ($x$ = 0, 0.05, 0.2).**
**a** Models of NiOOH with and without Fe dopant to show the Ni-O bond extension along z axis. **b** The calculated $e_g^*$ band Ni partial density of states (PDOS) of Ni$_{1-x}$Fe$_x$OOH ($x$ = 0, 0.05, 0,2). **c** Raman spectra of the Ni$_{1-x}$Fe$_x$OOH ($x$ = 0, 0.05, 0,2) to compare the intensity ratio between peak at 472 cm$^{-1}$ and 555 cm$^{-1}$ ($I_{472}/I_{555}$). **d** FT-EXAFS spectra of Ni $K$ edges of Ni$_{1-x}$Fe$_x$(OH)$_2$ (x = 0, 0.05, 0.2), inset showing the enlarged FT results within the range of 1.5 to 1.75 Å. **e** The correlation between the Ni $K$-edge white line intensity with the energy difference of $d_{x^2-y^2}$ and $d_{z^2}$ orbitals for Ni$_{1-x}$Fe$_x$OOH ($x$ = 0, 0.05, 0.2). It should be noted that the white line intensity of Ni$_{1-x}$Fe$_x$(OH)$_2$ (x = 0, 0.05, 0.2) extracted from Fig. 5a in our previous work[20] is used to evaluate the $e_g^*$ band broadening for Ni$_{1-x}$Fe$_x$OOH (x = 0, 0.05, 0.2) to avoid the possible effects on the XAS results caused by the self-discharge of Ni$^{3+}$[31].

could be induced by both cation dopants and strain effect[20]. Hence, to validate the proposed hypothesis, it is necessary to verify this in systems influenced by strain effects. Specifically, the reconstruction derived X-NiOOH (X = NiS$_2$, NiSe$_2$, and Ni$_5$P$_4$) samples are prepared through the electro-oxidation of NiS$_2$/NiSe$_2$/Ni$_5$P$_4$ at a current density of 10 mA cm$^{-2}$ for 10 h, which would exhibit different extent of $e_g^*$ band broadening due to the strain effect, with the following order: NiS$_2$-NiOOH>NiSe$_2$-NiOOH >Ni$_5$P$_4$-NiOOH >NiOOH[20]. Here, the OER

activities of X-NiOOH (X = NiS$_2$, NiSe$_2$, Ni$_5$P$_4$) are investigated under both light and dark conditions. As shown in Fig. 5a, discernible discrepancies could be observed in the degree of activity enhancement among X-NiOOH (X = NiS$_2$, NiSe$_2$, Ni$_5$P$_4$) between light and dark conditions. NiS$_2$-NiOOH displays the most substantial improvement in OER activity under light irradiation, with an overpotential drop of 23 mV at 10 mA cm$^{-2}$. In comparison, NiSe$_2$-NiOOH exhibits a lower degree of OER activity promotion, with a 15 mV reduction in

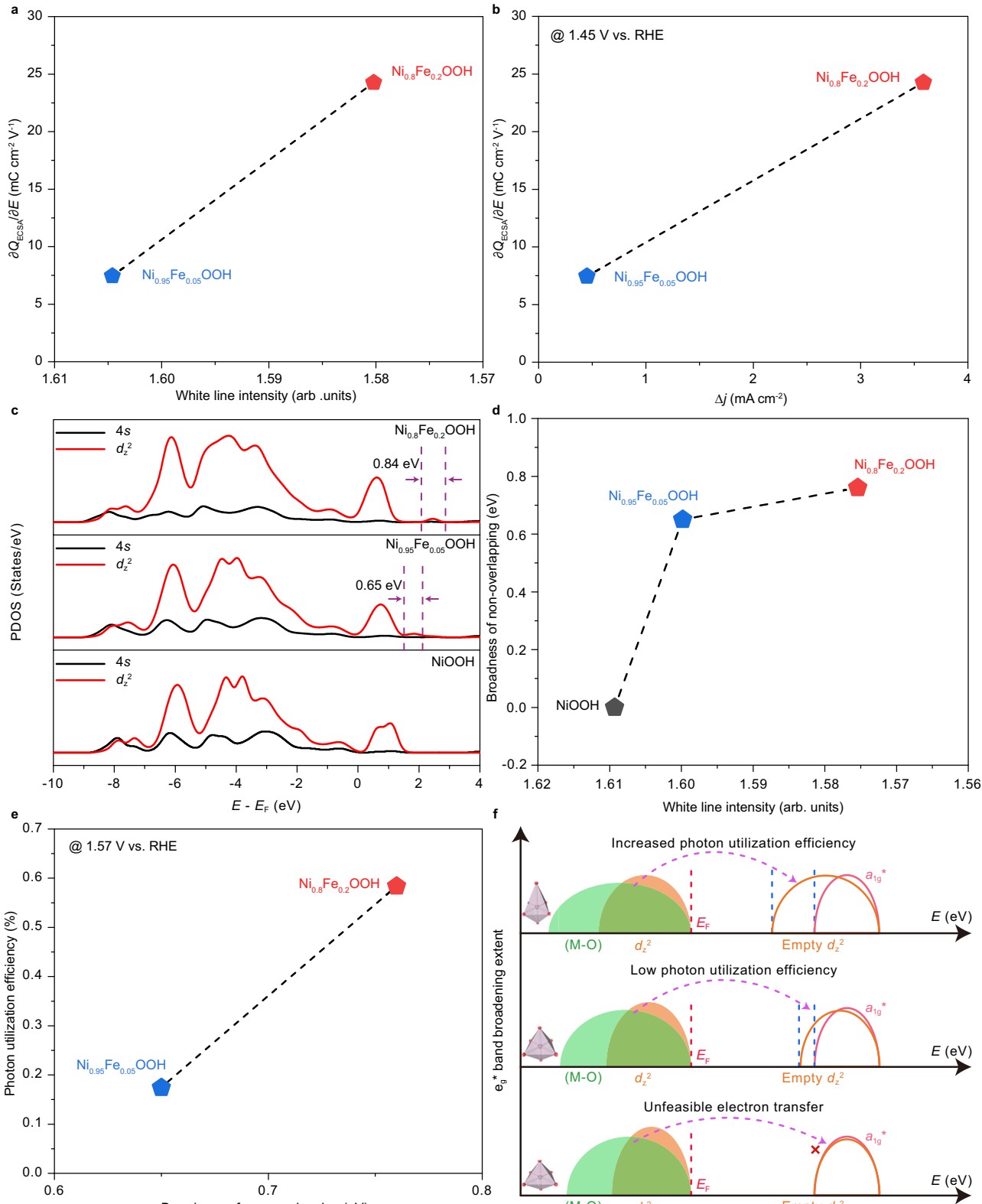

**Fig. 4 | Effects of $e_g^*$ band broadening on COM. a** The correlation between the Ni $K$-edge white line intensity with the $^{\cdot}$OH deprotonation ability of $Ni_{1-x}Fe_xOOH$ (x = 0.05, 0.2). **b** The correlation between the $^{\cdot}$OH deprotonation ability with the COM-contributed current density improvement of $Ni_{1-x}Fe_xOOH$ (x = 0.05, 0.2) under light condition at 1.45 V vs. RHE. **c** The calculated Ni PDOS of 4 $s$ and $d_z^2$ orbitals of $Ni_{1-x}Fe_xOOH$ (x = 0, 0.05, 0.2). **d** The correlation between the broadness of non-overlapping region and the Ni $K$-edge white line intensity of $Ni_{1-x}Fe_xOOH$ (x = 0, 0.05, 0.2). **e** The correlation between the broadness of non-overlapping region and the photon utilization efficiency of $Ni_{1-x}Fe_xOOH$ (x = 0.05, 0.2). **f** Schematic illustration of the effects of $e_g^*$ band broadening on facilitating the electron transfer from (M-O) to empty $d_z^2$ to increase the photon utilization efficiency under COM route.

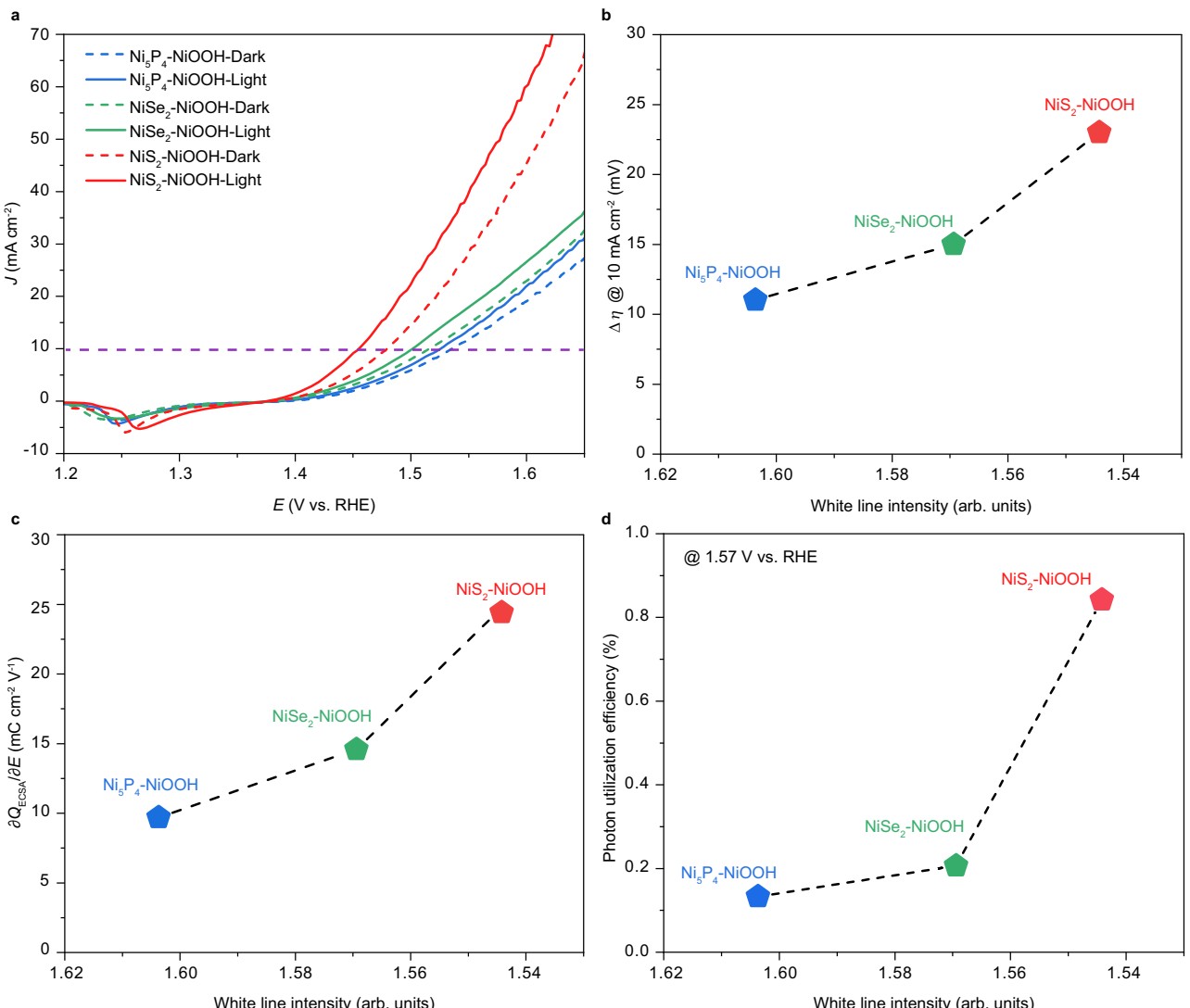

**Fig. 5 | Effects of increasing $e_g^*$ band broadening extent on COM-based OER activity of X-NiOOH (X = NiS₂, NiSe₂, and Ni₅P₄).** **a** LSV polarization curves of X-NiOOH (X = NiS₂, NiSe₂, and Ni₅P₄) based on a backward scan conducted at a scan rate of 0.1 mV s⁻¹.(without *iR*-correction) **b** The correlation between the COM contributed overpotential drop at 10 mA cm⁻² and the Ni *K*-edge white line intensity of X-NiOOH (X = NiS₂, NiSe₂, and Ni₅P₄). **c** The correlation between the Ni *K*-edge white line intensity with the *OH deprotonation ability of NiOOH (X = NiS₂, NiSe₂, and Ni₅P₄). **d** The correlation between the Ni *K*-edge white line intensity and the photon utilization efficiency of X-NiOOH (X = NiS₂, NiSe₂, and Ni₅P₄). It should be noted that the white line intensity of X-Ni(OH)₂ (X = NiS₂, NiSe₂, Ni₅P₄) is extracted from Fig. 3a in our previous work[20] to evaluate the $e_g^*$ band broadening for X-NiOOH to avoid the possible effects on the XAS results caused by the self-discharge of Ni³⁺[31].

overpotential, but a higher promotion compared to Ni₅P₄-NiOOH, which shows an 11 mV overpotential reduction. As such, a positive correlation between the extent of $e_g^*$ band broadening and the enhanced OER activities is revealed (Fig. 5b). At the same time, both the *OH deprotonation ability and photon utilization efficiency of X-NiOOH (X = NiS₂, NiSe₂, Ni₅P₄) follow the same order: NiS₂-NiOOH>NiSe₂-NiOOH >Ni₅P₄-NiOOH (Fig. 5c, d, Supplementary Fig. 14). These results demonstrate the universal concept that increased $e_g^*$ band broadening could simultaneously facilitate *OH deprotonation and promote photon utilization, resulting in highly efficient COM-based OER activity.

## Discussion

In this work, Ni₁₋ₓFeₓOOH (x = 0, 0.05, 0.2) samples are prepared to investigate the role of $e_g^*$ band broadening on enhancing the COM-based OER activity. The results demonstrate that a stronger $e_g^*$ band broadening would not only facilitate *OH deprotonation, but also promote photon utilization efficiency by generating a broader non-

overlapping region between $d_{z^2}$ and $a_{1g}^*$. As such, both of the two RDS in the COM route could be effectively alleviated by realizing stronger $e_g^*$ band broadening, resulting in higher OER activity. Furthermore, the universality of this concept is demonstrated by revealing the same effect on promoting the COM-based OER activity in X-NiOOH (X = NiS₂, NiSe₂, Ni₅P₄) with different $e_g^*$ band broadening extent. Previously, it was also unveiled that the stronger $e_g^*$ band broadening would trigger the emergence of more electronic states around the Fermi level. This effect notably facilitated the electron transfer from electrocatalysts to external circuit via *OH deprotonation, resulting in higher OER activities (AEM-pathway)[20]. As for the LOM route, the deprotonation step is usually the RDS. The conceptual framework encompassing NiO₆ distortion, $e_g^*$ band broadening, and *OH deprotonation also holds the potential for application in the context of the LOM process, thereby enabling the realization of enhanced catalytic performance. Therefore, it is believed that increasing the degree of $e_g^*$ band broadening could be a crucial factor on promoting the participation of COM under light

irradiation and even provide valuable guidelines for future design of highly efficient OER catalysts following various reaction pathways.

# Methods

## Materials
Nickel nitrate hexahydrate ($Ni(NO)_3 \cdot 6H_2O$), iron nitrate nonahydrate ($Fe(NO_3)_3 \cdot 9H_2O$), Urea, Sulfur, Selenium, Sodium phosphate monobasic were purchased from the Sigma-Aldrich. These chemicals were reagent grade and used as received without further purification.

## Synthesis of the $Ni_{1-x}Fe_x(OH)_2$ (x = 0, 0.05, 0,2)
The $Ni_{1-x}Fe_x(OH)_2$ (x = 0, 0.05, 0,2) were grown on the carbon cloth by a facile hydrothermal treatment[20]. Before hydrothermal treatment, the carbon cloth was pre-treated under 500 °C for 1 h in the air condition and then dealt with Ultra-Violet Ozone device for 30 min to make it fully hydrophilic. Next, 2 mmol total amount of $Ni(NO)_3 \cdot 6H_2O$ and $Fe(NO_3)_3 \cdot 9H_2O$, 10 mmol urea were added into 35 mL deionized (DI) water and stirred for 15 min to form uniform solution. One piece of carbon cloth with size of 2 cm × 3 cm was immersed into the solution. Then, they were transferred into a 50 mL Teflon-lined stainless-steel autoclave and kept in oven at 120 °C for 10 h. The obtained $Ni_{1-x}Fe_x(OH)_2$ (x = 0, 0.05, 0,2) was then washed by DI water and ethanol for at least three times, and further dried under 70 °C in air for 4 h.

## Synthesis of the X-NiOOH (X = $NiS_2$, $NiSe_2$, $Ni_5P_4$)
The X-NiOOH (X = $NiS_2$, $NiSe_2$, $Ni_5P_4$) was obtained from the reconstruction of $NiS_2$, $NiSe_2$, and $Ni_5P_4$ via a 10 h chronopotentiometry treatment[20]. First, one piece of hydrothermal-derived $Ni(OH)_2$ loaded carbon cloth was placed at the downstream position in crucible, while 400 mg sulfur/selenium/sodium phosphate monobasic power was placed at the upstream position. Then, the crucible was kept at 400 °C for 2 h under $N_2$ atmosphere and then cooled down to temperature. After that, the synthesized $NiS_2/NiSe_2/Ni_5P_4$ was washed using DI water and ethanol for at least three times, and dried at 70 °C for 1 h. Next, the chronopotentiometry treatment was conducted under 10 mA $cm^{-2}$ for 10 h to fully transfer the $NiS_2/NiSe_2/Ni_5P_4$ to X-NiOOH (X = $NiS_2$, $NiSe_2$, and $Ni_5P_4$).

## Removal of Fe impurity
The 1 M KOH solution was purified to remove Fe impurity before electrochemical measurement. To be specific, 0.5 g $Ni(NO)_3 \cdot 6H_2O$ powder was added into 30 mL 1 M KOH solution, forming $Ni(OH)_2$ precipitation. Next, the suspension was centrifuged at 7826 × g for 10 minutes. Then, the obtained $Ni(OH)_2$ powder was added into 50 mL KOH solution and mechanically agitated for 10 min. After standing still for 24 h, the suspension was centrifuged at 7826 × g for 10 minutes, and the KOH supernatant was kept in a clean electrochemical cell for use. The pH value of the purified 1 M KOH is detected to be 13.65 and the error bars represent mean ± standard error (Supplementary Fig. 2).

## Material characterizations
The Raman spectra were recorded by a Raman Spectrophotometer with an excitation wavelength of 514.4 nm. Nickel K-edge X-ray absorption fine structure (XAFS) spectra were recorded at the XAFCA beamline at the Singapore Synchrotron Light Source (SSLS) under transmission mode where the storage ring is running at 0.7 GeV with current nearly 200 mA[24]. The energy calibrations were finished by using standard Nickel foil. The $k^2$-weighted Fourier transforms were conducted using the Hanning window function for the EXAFS results, with the k-range of 2.5–10.5 $Å^{-1}$.

## Electrochemical measurements
All the electrochemical measurements were performed using an electrochemical workstation (VPM3, BiO-logic Inc) in a three-electrode setup in 1 M KOH. The working electrode was the $Ni_{1-x}Fe_xOOH$ (x = 0, 0.05, 0,2) and X-NiOOH (X = $NiS_2$, $NiSe_2$, $Ni_5P_4$). The Hg/HgO was chosen as the reference electrode. The Pt was used as the counter electrode. The linear scan voltammetry (LSV) was measured at scan rate 0.1 mV $s^{-1}$. In the dark situation, all electrochemical measurements are conducted in a dark box. Under light condition, a solar simulator (NBeT Solar-500, 300 W) was used as the light source to match AM 1.5 G. The intensity of the light irradiated on the electrocatalyst was 100 mW $cm^{-2}$ as calibrated by a light power meter (Newport 843-R). The experimental set-up is provided in Supplementary Fig. 15. Before evaluating the OER activity, all samples were subjected to chronopotentiometry measurements at a current density of 10 mA $cm^{-2}$ under dark for 24 h to ensure the complete oxidation of $Ni^{2+}$ to $Ni^{3+}$.

## Pulse-voltammetry measurement
One small piece of electrocatalysts on carbon cloth (0.2 $cm^2$) after electrochemical oxidation was used for the pulse-voltammetry measurement. The potential was set firstly at a low potential ($E_l$ = 1.40 V) for 80 s to stabilize the electrode surface. Then the potential was turned to a higher potential ($E_h$ = 1.42 V) for 12 s and back to $E_l$ for 12 s as one cycle. This cycle was repeated while increasing $E_h$ from 1.42 V to 1.48 V versus RHE in 20 mV/step with constant $E_l$ = 1.40 V. The transferred charge normalized to ECSA during each cycle was evaluated by integrating the current pulse/ECSA over time.

## Electrochemically surface area
The electrochemically surface area (ECSA) data was evaluated from recording the electrochemical double-layer capacitance of the catalyst via cyclic voltammograms (CVs). Here, the potential range was set at 0.02–0.12 V (versus Hg/HgO) to avoid the Faradaic process. The CVs were conducted in the quiescent electrolyte with the potential swept across the set potential range with at 6 scan rates 5, 10, 20, 30, 40, 50 mV $s^{-1}$. The charging current was plotted versus scan rate and a straight line could be derived with the slope value equalled to the double-layer capacitance ($C_{dl}$). The ECSA was obtained by dividing the $C_{dl}$ to the specific capacitance $C_s$ = 0.04 mF $cm^{-2}$ [25].

## Photon utilization calculation
In COM route, two photons participated into one oxygen molecule evolution, which corresponded to four electrons. Hence, the utilized photon number was calculated by the equation:

$$N_0 = 0.5 \times \Delta J \times t \times S / e \tag{1}$$

in which $\Delta J$ was the current density difference between light and dark conditions at 1.57 V vs. RHE, $t$ was reaction time, $S$ was surface area, e was elementary charge. The number of photons per second and surface unit for certain wavelength was calculated by

$$N_p = I/E_p = I \times \lambda / hc = I \times \lambda \times 10^{-9} / (1.988 \times 10^{-25})$$
$$= I \times \lambda \times 5.03 \times 10^{15} [\text{Photons m}^{-2} \text{s}^{-1} \text{nm}^{-1}] \tag{2}$$

where $I$ [W $m^{-2}$ $nm^{-1}$] was the irradiance for certain wavelength, which could be known from Supplementary Fig. 5a; $E_P$ was the photon energy, h was Planck constant, c was speed of light, $\lambda$ was wavelength. The $N_P$-$\lambda$ spectra could therefore be obtained, as shown in Supplementary Fig. 5b. Integration of the $N_P$-$\lambda$ spectra yields a theoretical maximum photon flux $N$ of $4.936 \times 10^{22}$ photons $m^{-2}$ $s^{-1}$. Hence, the photon utilization efficiency was calculated as

$$N_0/N_{\text{Total}} = (0.5\Delta J \times t \times S/e)/(N \times t \times S) = 0.5\Delta J/(N \times e) \tag{3}$$

## Computational method

All calculations were conducted via the DFT with the generalized Perdew-Burke-Ernzerhof (PBE)[26], and the projector augmented-wave (PAW)[27] pseudopotential planewave method as implemented in the Vienna ab initio Simulation Package (VASP) code[28]. For the PAW pseudopotential, $3d^84s^2$, $3d^74s^1$, $2s^22p^4$ and $1s^1$ were treated as valence electrons for Ni, Fe, O and H atoms, respectively. The bulk β-NiOOH structure was optimized within the local-spin-polarized density approximations (LSDA + U, $U_{eff}$ = 5.3 eV for Ni and 5 eV for Fe, respectively) .The plane wave cutoff energy of 500 eV was set for all calculations and the energy and force convergence were set to $10^{-5}$ eV and 0.02 eV Å$^{-1}$, respectively. Grimme's DFT-D3(BJ) dispersion correction[29] were used in all calculations. A 12 × 12 × 10 Monkhorst-Pack (MP) K-point grid was used for β-NiOOH unit cells geometry optimization calculations. The layered NiOOH structure was optimized using the same basic settings of the bulk β-NiOOH structure optimization except the k-point sampling (12 × 6 × 1). The optimized unit cell of layered NiOOH was expanded to 6 × 3 supercells (a = 17.556 Å and b = 17.739 Å) containing 72 Ni, 72 H and 144 O atoms with a vacuum thickness of 18 Å for Fe embedding systems. Based on our experimental results, two different Fe doping concentrations were considered in this study, they are $Ni_{0.95}Fe_{0.05}OOH$ and $Ni_{0.8}Fe_{0.2}OOH$ corresponding to computational models of $Fe_4Ni_{68}$ and $Fe_{14}Ni_{58}$, respectively (Supplementary Fig. 7, the CONTCAR of the optimized $Ni_{1-x}Fe_xOOH$ (x = 0, 0.05, 0.2) models is provided in Supplementary Data 1). It should be noted that we constrained lattice constants and relaxed all atoms in Fe embedding systems simulations. For supercell case, only γ K-point was considered.

For COM route, the thermodynamic correction of Gibbs free energy was implemented by VASPKIT[30] and the relative energy was calculated by:

$$\Delta G(1) = G(O^*O) + \frac{G(H_2)}{2} - G(O^*OH) - eU \quad (4)$$

$$\Delta G(2) = G(^*OO) - G(O^*O)(\text{Light dominated process}) \quad (5)$$

$$\Delta G(3) = G(HO^*O_2) + \frac{G(H_2)}{2} - G(^*OO) - G(H_2O) - eU \quad (6)$$

$$\Delta G(4) = G(HO^*OH) + G(O_2) + \frac{G(H_2)}{2} - G(HO^*O_2) - G(H_2O) - eU \quad (7)$$

$$\Delta G(5) = G(O^*OH) + \frac{G(H_2)}{2} - G(HO^*OH) - eU \quad (8)$$

## Data availability

All the data supporting of the finding of this study are included within the paper and its supporting files and are available from the corresponding authors on request.

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

## Acknowledgements

The authors gratefully acknowledge the financial support provided by the National Research Foundation, Singapore, under its Competitive Research Programme (Award No.: NRF-CRP26-2021-0003) and Ministry of Education, Singapore, under its Academic Research Fund (AcRF) Tier 2 (Award No: MOE-T2EP501220010). The computational resource was provided by A*STAR Computational Resource Centre, Singapore (A*CRC) and the National Supercomputing Centre, Singapore (https://www.nscc.sg). Y-W. Z acknowledges the support from A*STAR-SERC-CRF Award.

## Author contributions

H.Y.Z., X.P.W., S.B.X., J.Z., and J.M.X. conceived the idea. H.Y.Z. and X.P.W. performed synthesis and electrochemical measurement of the samples. H.Y.Z., X.P.W., J.C.Y, X.Z, H.A., Q.Z., Y.F.M., H.W., were responsible for the analysis of electrochemical results. S.B.X, C.W., C.Z.D., were responsible for the XAFS characterization. Y-W. Z., Z.G.Y., and Q.Z. carried out DFT simulations. J.M.X. is in charge of the overall project and preparation of the manuscript.

## Competing interests

The authors declare no competing interests.
