## [Peer Review File · Nature Communications]

REVIEWER COMMENTS

Reviewer #1 (Remarks to the Author):

In this manuscript, the authors reveal the relationship between the extent of eg^* band broadening and enhanced OER activities based on COM route with using $Ni_{1-x}Fe_xOOH$ ($x = 0, 0.05, 0.2$) as the model catalysts. This work might be a further motivation for electronical engineering strategy for OER, but it needs a careful revision before further considerations.

1. During the OER measurement, the surface corrosion or reconstruction is generally carried out. So what about this situation for this catalyst. More detailed characterizations are needed.
2. OER performance of $Ni_{1-x}Fe_xOOH$ ($x = 0, 0.05, 0.2$) at light with different wavelength needs to be measured to further confirm this effect.
3. Some important data are missing. For example, in Figure 5b-d, the spectra data of Ni K-edge of $NiOOH$ ($X = NiS_2, NiSe_2, and Ni_5P_4$).

Reviewer #2 (Remarks to the Author):

In this work, the authors proposed deeper insights to their previously published coupled oxygen evolution mechanism (COM). The relationship between the extent of eg^* band broadening and enhanced OER activities is elaborated upon in detail. The improvement of COM helps researchers to understand the OER process from a new perspective beyond existing AEM and LOM processes. It is an insightful work and the reported fruitful results are of fundamental interests for the research community. The manuscript has been properly organized as well. The description, expression as well as target of language is very clear. However, there are certain aspects that need to be clarified and some of them can influence the overall quality of the work. below are the concerns:

1. What is the origin of the negative current at about 1.25 V in Fig 1a? Why it only happens to $NiOOH$ rather than Fe doped $Ni_{1-x}Fe_xOOH$?
2. In Fig S3, the proportion of Fe atoms does not seem to match 0.05 and 0.2. Please give more details like lattice parameters, number of atoms.

3. As for the band center difference between $d_{x^2-y^2}$ and d_{z^2} , the authors pointed out the greater e_g^* band broadening extent induced by high concentration of Fe dopants. However, the catalytic performance is generally enhanced with the increasing d-band center according to the d-band center theory. In this work, I found the d-band center exhibits a downward trend. How to understand the differences between these two theories?

4. Please give more details of computational method. Did the authors take spin polarized calculation in their simulations? How about the set of cutoff energy, convergence criterion and k-mesh?

5. How to understand the COM mechanism from a theoretical perspective? The authors described "The deprotonation of *OH is quite similar to that in the AEM route", can people use COM mechanism to draw a step diagram like AEM or LOM? For example, first step, $^*OH \rightarrow$ second step, $\dots \rightarrow$ xxx step, O_2 .

6. The definition of x in $Ni_{1-x}Fe_xOOH$ is unclear, e.g., in line 67, 85 is " $Ni_{1-x}Fe_xOOH$ ($x = 0, 0.05, 0.2$)" while line 105 is " $Ni_xFe_{1-x}OOH$ ($0, 0.05, 0.2$)", which is correct?

7. Why did you choose x in this range ($x = 0, 0.05, 0.2$)/ ($0, 0.05, 0.2$)? Is there a special reason? What happens when $x=1$?

8. How to distinguish whether the effect on broadening of the e_g^* band of $Ni_{1-x}Fe_xOOH$ is Fe doping or strain effect?

Reviewer #3 (Remarks to the Author):

In this work, the authors systematically investigate the coupled oxygen evolution mechanism (COM) in $NiFeOOH$ model catalysts. According to their previous study, the light-triggered COM pathway could promote OER activity significantly by realizing both metal and oxygen redox, provided that a unit cell configuration, e.g. NiO_6 octahedral units in $NiOOH$ structure experience sufficient distortion to have modulated electronic states with both metal and oxygen bands around fermi level. In this study, the authors decipher the relationship between e_g^* band broadening and OER activity enhancement. Such correlation was investigated at two regions of low and high OER potential, where at low potentials e_g^* band broadening expedites *OH deprotonation as the RDS, while at high potentials it promotes the non-overlapping region between d_{z^2} and a_{1g}^* states which result in higher photon utilization efficiency. By conducting systematic electrochemical measurements with Fe-doped $NiOOH$ catalysts showing varying degrees of e_g^* band broadening and thus NiO_6 distortion, Raman and XAS spectroscopies, and DFT calculations, the facilitated proton transfer at *OH deprotonation RDS at low potential as well as light-triggered electron transfer from M-O bands to d_{z^2} orbitals are elucidated. The established relationships between tuned electronic states and improved OER at both low and high potentials are demonstrated for more complex structures to show the universal applicability of COM in OER catalyst development. Overall, this work is novel with significant analyses and discussion, thanks to a suite of advanced

measurements and modeling. I would recommend accepting this work after revision of some issues listed below.

1- In page 3, Fig 1a is described as “the highest OER activity is observed for Ni_{0.8}Fe_{0.2}OOH under dark condition”, while it is the opposite in the figure (light condition).

2- The pulse-voltammetry measurements and the corresponded slopes for the understudy structures are missing. Please provide the measurements and calculated slopes.

3- Could the authors discuss over the reported findings and the concept of NiO₆ distortion, *OH deprotonation, and eg* band broadening in normal ambient electrolysis conditions? If no light/dark conditions are applied, would the found relationships happen at all and thus impact AEM or even LOM pathways for promoting OER?

4- The authors should elucidate and discuss why necessarily the established relationships between eg* band broadening, *OH deprotonation, and photon utilization promote OER for sulfur compared to selenium and phosphorous. Why having NiS₂ interfaced with NiOOH would instigate such correlations?

5- The language should be improved.

Response to referees

We thank the three referees for taking the time to carefully review the manuscript and for giving many useful suggestions. The manuscript has been revised according to their comments and the changes have been highlighted in the revised version. We feel the quality of the paper has been improved greatly thanks to the input from the referees. Below is a point-by-point response.

Reviewer #1

Remarks to the Author: In this manuscript, the authors reveal the relationship between the extent of eg* band broadening and enhanced OER activities based on COM route with using Ni_{1-x}Fe_xOOH (x= 0, 0.05, 0.2) as the model catalysts. This work might be a further motivation for electrochemical engineering strategy for OER, but it needs a careful revision before further considerations.

1. During the OER measurement, the surface corrosion or reconstruction is generally carried out. So what about this situation for this catalyst. More detailed characterizations are needed.

Response: We are very grateful for the referee's comment. We agree that the electrocatalysts might undergo surface reconstruction process or surface corrosion during the OER measurement. In our work, to study whether there was a surface reconstruction or surface corrosion during OER measurement, the samples before and after OER measurement was analyzed using Ni *K*-edge and *L*-edge XAS measurement. As shown in Fig.R1-1, there is negligible structural difference for the samples after OER under dark or light conditions. Therefore, it could be inferred that the light induced geometry conversion is highly reversible and would not lead to structure corrosion or reconstruction during OER process.

Fig.R1-1 Ni *K*-edge spectra (a) and *L*-edge spectra (b) of $\text{Ni}_{0.95}\text{Fe}_{0.05}(\text{OH})_2$ after OER under dark or light condition; Ni *K*-edge spectra (c) and *L*-edge spectra (d) of $\text{Ni}_{0.8}\text{Fe}_{0.2}(\text{OH})_2$ after OER under dark or light condition. Here, it should be noted that $\text{Ni}_{0.95}\text{Fe}_{0.05}\text{OOH}$ and $\text{Ni}_{0.8}\text{Fe}_{0.2}\text{OOH}$ samples were completely reduced to $\text{Ni}_{0.95}\text{Fe}_{0.05}(\text{OH})_2$ and $\text{Ni}_{0.8}\text{Fe}_{0.2}(\text{OH})_2$ using ethanol (following the reaction equation: $\text{Ni}^{3+} + \text{CH}_3\text{CH}_2\text{OH} = \text{Ni}^{2+} + \text{CH}_3\text{CHO}$, *Journal of The Electrochemical Society*, 2015, 163(2): H99.) to avoid the possible effects on the XAS results caused by the self-discharge of Ni^{3+} . (Fig.S10 in supplementary materials)

Moreover, in our prior research, we have demonstrated that the geometry conversion between NiO_6 and NiO_4 in nanoribbon- NiOOH (NR- NiOOH) under light condition is a reversible process. (*Nature*, 2022, 611(7937): 702-708.) As shown in Fig.R1-2 a, when the light source is switched off, a gradual blue shift in the absorption edge with increased white line intensity for the sample subjected to the light irradiation is observed. Furthermore, the intensities of both Ni-O and Ni-Ni bonds gradually increase. (Fig.R1-2 b) The obtained results suggest that the structure of the sample progressively returns to its initial configuration under dark conditions, providing confirmation of the high reversibility of the light-induced geometric conversion.

Fig.R1-2 Operando XAFS spectra of NR-NiOOH switching off light irradiation. (a) Operando Ni K-edge XANES spectra of NR-NiOOH switching off light irradiation (inset showing the electrochemical curves accompanying the operando-XAFS); (b) FT-EXAFS spectra of NR-NiOOH switching off light irradiation. Reproduced with permission. Copyright 2022, Springer Nature (Fig.S12, *Nature*, 2022, 611(7937): 702-708.)

As such, during the OER measurement, the surface corrosion or reconstruction did not occur. To address the comment, we have revised our manuscript as follows (line 206, page 8, highlighted in yellow): Moreover, both Ni *K*-edge and *L*-edge XAS measurements show that there is negligible structural difference for the samples after OER under dark or light conditions, confirming the reversibility of the electrode surface during the COM-based OER process. (Fig.S10)

2. OER performance of Ni_{1-x}Fe_xOOH (*x* = 0, 0.05, 0.2) at light with different wavelength needs to be measured to further confirm this effect.

Response: We are very grateful for the referee's comment. In our work, it is revealed that a stronger e_g^* band broadening would result in the broadening of the non-overlapping region between d_z^2 and a_{1g}^* electronic states, leading to a higher photon utilization efficiency. To further confirm this effect, we conduct LSV measurements of Ni_{0.95}Fe_{0.05}OOH and Ni_{0.8}Fe_{0.2}OOH under light with wavelength at 575 nm, 475 nm, and 365 nm, respectively. The photon utilization efficiency at different wavelength can be evaluated by comparing the current density change between dark condition and light condition at specific wavelength at high potential region. As shown in Fig.R1-3d, the Ni_{0.8}Fe_{0.2}OOH with stronger e_g^* band broadening exhibits higher current density promotion at 1.57 V vs. RHE between dark condition and light at each wavelength than Ni_{0.95}Fe_{0.05}OOH. These results hence further proved that a stronger e_g^* band broadening would lead to a higher photon utilization efficiency.

Fig.R1-3 (a) LSV polarization curves of $\text{Ni}_{0.95}\text{Fe}_{0.05}\text{OOH}$ under light with wavelength at 575 nm, 475 nm, and 365 nm, respectively; (b) LSV polarization curves of $\text{Ni}_{0.8}\text{Fe}_{0.2}\text{OOH}$ under light with wavelength at 575 nm, 475 nm, and 365 nm, respectively; (c) Comparison of the current density at 1.57 V vs. RHE between dark condition and light condition at different wavelength (575nm, 475nm, 365nm) for $\text{Ni}_{1-x}\text{Fe}_x\text{OOH}$ ($x = 0.05, 0.2$). (Fig.S11 in supplementary materials)

To address the comment, we have revised our manuscript as follows (line 236, page 8, highlighted in yellow):

To further confirm such effect, the LSV measurements of $\text{Ni}_{0.95}\text{Fe}_{0.05}\text{OOH}$ and $\text{Ni}_{0.8}\text{Fe}_{0.2}\text{OOH}$ under light with wavelength at 575 nm, 475 nm, and 365 nm, respectively are conducted. The $\text{Ni}_{0.8}\text{Fe}_{0.2}\text{OOH}$ with stronger e_g^* band broadening exhibits higher current density promotion at 1.57 V vs. RHE between dark and light at each wavelength than that of $\text{Ni}_{0.95}\text{Fe}_{0.05}\text{OOH}$. (Fig.S11)

3. Some important data are missing. For example, in Figure 5b-d, the spectra data of Ni K-edge of NiOOH ($X = \text{NiS}_2, \text{NiSe}_2, \text{and Ni}_5\text{P}_4$).

Response: We are very grateful for the referee's comment. The structure of $X\text{-NiOOH}$ ($X = \text{NiS}_2, \text{NiSe}_2, \text{and Ni}_5\text{P}_4$) has been well studied in our previous work (*Energy & Environmental Science*, 2023, 16(2): 641-652).

Here, it should be noted that Ni^{3+} is unstable, which would be gradually reduced into Ni^{2+} via a self-discharge process. To avoid the possible effects on the XAS results caused by the self-discharge of Ni^{3+} , all the NiOOH samples were completely reduced to $\text{Ni}(\text{OH})_2$ state using ethanol (following the reaction equation: $\text{Ni}^{3+} + \text{CH}_3\text{CH}_2\text{OH} = \text{Ni}^{2+} + \text{CH}_3\text{CHO}$, *Journal of The Electrochemical Society*, 2015, 163(2): H99.) As shown in Figure R1-4, the white line of these reconstruction-derived $\text{Ni}(\text{OH})_2$ became broader with lower intensities in the extent of $\text{Ni}(\text{OH})_2 < \text{Ni}_5\text{P}_4\text{-Ni}(\text{OH})_2 < \text{NiSe}_2\text{-Ni}(\text{OH})_2 < \text{NiS}_2\text{-Ni}(\text{OH})_2$. It should be noted that a more distorted NiO_6 octahedron was typically represented by a broader white line with lower intensity. As such, it was observed that the extent of NiO_6 octahedron distortion was significantly increased in the order of $\text{Ni}(\text{OH})_2 < \text{Ni}_5\text{P}_4\text{-Ni}(\text{OH})_2 < \text{NiSe}_2\text{-Ni}(\text{OH})_2 < \text{NiS}_2\text{-Ni}(\text{OH})_2$. (Fig.R1-3 b). The stronger NiO_6 distortion would theoretically lead to a higher extent of e_g^* broadening, which had also been proved in the O *K*-edge spectra. (Fig.R1-3 d and e).

In this work, the white line intensity value was extracted from the previous Ni *K*-edge spectra to study the relationship between e_g^* band broadening and COM-based OER activities in the reconstruction-derived X-NiOOH system.

Fig. R1-4. Structural characterization of X-Ni(OH)₂ (X = NiS₂, NiSe₂, Ni₅P₄). **a.** Normalized Ni *K*-edge XANES spectra of X-Ni(OH)₂ (X= NiS₂, NiSe₂, Ni₅P₄) with Ni(OH)₂ as the benchmark (inset showing the enlarged results within the range of 8344 to 8352 eV). **b.** White line intensity (obtained from Fig.3a) comparison for hydrothermal prepared Ni(OH)₂. Lower intensity suggests stronger NiO₆ distortion. **c.** FT-EXAFS spectra of Ni *K*-edges of X-Ni(OH)₂ (X= NiS₂, NiSe₂, Ni₅P₄) and Ni(OH)₂ (inset showing the enlarged FT results within the range of 1.4 to 1.85 Å). **d.** Normalized O *K*-edge XAS spectra of X-Ni(OH)₂ (X= NiS₂, NiSe₂, Ni₅P₄) and hydrothermal prepared Ni(OH)₂. **e.** Schematic diagram of the effect of NiO₆ octahedron distortion on *e_g*^{*} band broadening for X-Ni(OH)₂ (X= NiS₂, NiSe₂, Ni₅P₄) with standard Ni(OH)₂ (no distortion) as the benchmark. Reproduced with permission. Copyright 2023, The Royal Society of Chemistry. (*Energy & Environmental Science*, 2023, 16(2): 641-652.)

To address the comment, we have revised our manuscript as follows

(line 286, page 11, highlighted in yellow): It should be noted that the white line intensity of X-Ni(OH)₂ (X = NiS₂, NiSe₂, Ni₅P₄) is extracted from Fig.3a in our previous work²² to evaluate the *e_g*^{*} band broadening for

X-NiOOH to avoid the possible effects on the XAS results caused by the self-discharge of Ni³⁺.²⁴

Reviewer #2

Remarks to the Author: In this work, the authors proposed deeper insights to their previously published coupled oxygen evolution mechanism (COM). The relationship between the extent of eg* band broadening and enhanced OER activities is elaborated upon in detail. The improvement of COM helps researchers to understand the OER process from a new perspective beyond existing AEM and LOM processes. It is an insightful work and the reported fruitful results are of fundamental interests for the research community. The manuscript has been properly organized as well. The description, expression as well as target of language is very clear. However, there are certain aspects that need to be clarified and some of them can influence the overall quality of the work. below are the concerns:

1. What is the origin of the negative current at about 1.25 V in Fig 1a? Why it only happens to NiOOH rather than Fe doped Ni_{1-x}Fe_xOOH?

Response: We are very grateful for the referee's comment. In our work, we conducted negative scan for the LSV measurement to avoid the influence of Ni^{2+/3+} redox current on the OER activity. The negative current at about 1.25 V can be ascribed to the redox from Ni³⁺ to Ni²⁺. This can also be observed for Ni_{1-x}Fe_xOOH ($x = 0.05, 0.2$) as shown in the enlarged Fig. 1a (Fig.R2-1).

Fig.R2-1 Enlarged LSV polarization curves of Ni_{1-x}Fe_xOOH ($x = 0, 0.05, 0.2$) from Fig.1a based on a backward scan conducted at a scan rate of 0.1 mV s⁻¹. (Fig.S2 in supplementary materials)

To address the comment, we have revised our manuscript as follows (line 92, page 3, highlighted in yellow):

Negative scan is conducted for the LSV measurement to avoid the influence of $\text{Ni}^{2+/3+}$ redox current on the OER activity. Hence the negative current peak due to the redox from Ni^{3+} to Ni^{2+} are observed for $\text{Ni}_{1-x}\text{Fe}_x\text{OOH}$ ($x = 0, 0.05, 0.2$). (Fig.1a and Fig.S2)

2. In Fig S3, the proportion of Fe atoms does not seem to match 0.05 and 0.2. Please give more details like lattice parameters, number of atoms.

Response: We are very grateful for the referee's comment. After carefully checked the Fe/Ni atom ratios in the simulation models, we find the Fe/(Fe+Ni) ratio for $\text{Ni}_{0.95}\text{Fe}_{0.05}\text{OOH}$ is 0.05, while the Fe/(Fe+Ni) ratio for $\text{Ni}_{0.8}\text{Fe}_{0.2}\text{OOH}$ is 0.16. Therefore, we reconduct the DFT calculations for $\text{Ni}_{0.8}\text{Fe}_{0.2}\text{OOH}$. The details like lattice parameters, number of atoms are provided as follows:

The layered NiOOH structure was optimized using the same basic settings of the bulk β -NiOOH structure optimization except the k-point sampling ($12 \times 6 \times 1$). The optimized unit cell of layered NiOOH was expanded to 6×3 supercells ($a=17.556 \text{ \AA}$ and $b=17.739 \text{ \AA}$) containing 72 Ni, 72 H and 144 O atoms with a vacuum thickness of 18 \AA for Fe embedding systems. Based on our experimental results, two different Fe doping concentrations were considered in this study, they are $\text{Ni}_{0.95}\text{Fe}_{0.05}\text{OOH}$ and $\text{Ni}_{0.8}\text{Fe}_{0.2}\text{OOH}$ corresponding to computational models of $\text{Fe}_4\text{Ni}_{68}$ and $\text{Fe}_{14}\text{Ni}_{58}$, respectively. (Fig.R2-2 a-c) It should be noted that we constrained lattice constants and relaxed all atoms in Fe embedding systems simulations. For supercell case, only γ k-point was considered.

Fig.R2-2. Simulation models of (a) NiOOH. (b) Ni_{0.95}Fe_{0.05}OOH. (c) Ni_{0.8}Fe_{0.2}OOH. The optimized unit cell of layered NiOOH was expanded to 6×3 supercells (a=17.556 Å and b=17.739 Å) containing 72 Ni, 72 H and 144 O atoms with a vacuum thickness of 18 Å for Fe embedding systems. The simulation results for Ni_{0.95}Fe_{0.05}OOH and Ni_{0.8}Fe_{0.2}OOH are corresponded to computational models of Fe₄Ni₆₈ and Fe₁₄Ni₅₈, respectively. (Fig.S6 in supplementary materials)

To address the comment, we have revised our manuscript as follows

(line 385, page 13, highlighted in yellow): The layered NiOOH structure was optimized using the same basic settings of the bulk β-NiOOH structure optimization except the k-point sampling (12×6×1). The optimized unit cell of layered NiOOH was expanded to 6×3 supercells (a=17.556 Å and b=17.739 Å) containing 72 Ni, 72 H and 144 O atoms with a vacuum thickness of 18 Å for Fe embedding systems. Based on our experimental results, two different Fe doping concentrations were considered in this study, they are Ni_{0.95}Fe_{0.05}OOH and Ni_{0.8}Fe_{0.2}OOH corresponding to computational models of Fe₄Ni₆₈ and Fe₁₄Ni₅₈, respectively. (Fig.S6 a-c) It should be noted that we constrained lattice constants and relaxed all atoms in Fe embedding systems simulations. For supercell case, only γ k-point was considered.

(Fig.S6, highlighted in yellow):

Fig. S6. Simulation models of (a) NiOOH. (b) Ni_{0.95}Fe_{0.05}OOH. (c) Ni_{0.8}Fe_{0.2}OOH. The optimized unit cell of layered NiOOH was expanded to 6×3 supercells (a=17.556 Å and b=17.739 Å) containing 72 Ni, 72 H and 144 O atoms with a vacuum thickness of 18 Å for Fe embedding systems. The simulation results for Ni_{0.95}Fe_{0.05}OOH and Ni_{0.8}Fe_{0.2}OOH are corresponded to computational models of Fe₄Ni₆₈ and Fe₁₄Ni₅₈, respectively.

Besides, the optimized POSCARs (CONTCARs) for Ni_{1-x}Fe_xOOH (x=0,0.05,0.2) are also provided. (Appendix part of Supplementary Materials)

3. As for the band center difference between dx²-y² and dz², the authors pointed out the greater eg* band broadening extent induced by high concentration of Fe dopants. However, the catalytic performance is generally enhanced with the increasing d-band center according to the d-band center theory. In this work, I found the d-band center exhibits a downward trend. How to understand the differences between these two

theories?

Response: We are very grateful for the referee's comment. The d-band theory is mainly proposed for understanding the bond formation and trends in reactivity among transition metals, which is an approximate description of the bond formation at a transition metal surface. (*Proceedings of the National Academy of Sciences*, 2011, 108(3): 937-943.) Hence, the d-band theory has been widely used as a descriptor in the hydrogen evolution reaction, where the adsorption of atoms/molecules onto the transition metal surfaces is very important for catalytic performance.

However, for OER, the catalytic performance might not be enhanced with the increasing d-band center, as shown in Figure R2-3. (*ACS Catal.* 10(16), 9086–9097 (2020).) At the same time, in our work the catalytic materials for OER were metal oxyhydroxides. As compared with the transition metals, the electronic structure of metal cations would undergo significant changes upon being surrounded by an array of point charges generated by oxygen ions. Hence, it is not straightforward to describe the OER activities of transition metal oxyhydroxides by means of d-band theory. For metal oxyhydroxides, the catalytic performance is significantly influenced by the electronic states around the Fermi level, i.e., e_g^* band for NiOOH species. As the $d_{x^2-y^2}$ and d_{z^2} becomes more splitting, more electronic states would be around the Fermi level, hence significantly facilitating the electron transfer, leading to highly efficient OER activities. (*Energy & Environmental Science*, 2023, 16(2): 641-652.; *Nature Communications*, 2019, 10(1): 2713.)

Fig.R2-3 Relationships of the calculated d-band centers and experimental overpotential of Fe-doped Ni₂P

4. Please give more details of computational method. Did the authors take spin polarized calculation in their simulations? How about the set of cutoff energy, convergence criterion and k-mesh?

Response: We are very grateful for the referee's comment. All calculations were conducted via the DFT with the generalized Perdew-Burke-Ernzerhof (PBE), and the projector augmented-wave (PAW) pseudopotential plane wave method as implemented in the Vienna ab initio Simulation Package (VASP) code. For the PAW pseudopotential, $3d^8 4s^2$, $3d^7 4s^1$, $2s^2 2p^4$ and $1s^1$ were treated as valence electrons for Ni, Fe, O and H atoms, respectively. The bulk β -NiOOH structure was optimized within the local-spin-polarized density approximations (LSDA+U, $U_{\text{eff}}=5.3$ eV for Ni and 5 eV for Fe, respectively). The plane wave cutoff energy of 500 eV was set for all calculations and the energy and force convergence were set to 10^{-5} eV and 0.02 eV \AA^{-1} , respectively. Grimme's DFT-D3(BJ) dispersion correction were used in all calculations. A $12 \times 12 \times 10$ Monkhorst-Pack (MP) k-point grid was used for β -NiOOH unit cells geometry optimization calculations. The layered NiOOH structure was optimized using the same basic settings of the bulk β -NiOOH structure optimization except the k-point sampling ($12 \times 6 \times 1$). The optimized unit cell of layered NiOOH was expanded to 6×3 supercells ($a=17.556$ \AA and $b=17.739$ \AA) containing 72 Ni, 72 H and 144 O atoms with a vacuum thickness of 18 \AA for Fe embedding systems. Based on our experimental results, two different Fe doping concentrations were considered in this study, they are $\text{Ni}_{0.95}\text{Fe}_{0.05}\text{OOH}$ and $\text{Ni}_{0.8}\text{Fe}_{0.2}\text{OOH}$ corresponding to computational models of $\text{Fe}_4\text{Ni}_{68}$ and $\text{Fe}_{14}\text{Ni}_{58}$, respectively. (Fig.S6 a-c) It should be noted that we constrained lattice constants and relaxed all atoms in Fe embedding systems simulations. For supercell case, only γ k-point was considered.

For COM route, the thermodynamic correction of Gibbs free energy was implemented by VASPKIT and the relative energy was calculated by:

$$\Delta G(1) = G(\text{O}^*\text{O}) + \frac{G(\text{H}_2)}{2} - G(\text{O}^*\text{OH}) - eU$$

$$\Delta G(2) = G(^*\text{OO}) - G(\text{O}^*\text{O}) \quad (\text{Light dominated process})$$

$$\Delta G(3)=G(\text{HO}^*\text{O}_2)+\frac{G(\text{H}_2)}{2}-G(^*\text{OO})-G(\text{H}_2\text{O})-eU$$

$$\Delta G(4)=G(\text{HO}^*\text{OH})+G(\text{O}_2)+\frac{G(\text{H}_2)}{2}-G(\text{HO}^*\text{O}_2)-G(\text{H}_2\text{O})-eU$$

$$\Delta G(5)=G(\text{O}^*\text{OH})+\frac{G(\text{H}_2)}{2}-G(\text{HO}^*\text{OH})-eU$$

To address the comment, we have revised our manuscript as follows (line 377, page 13, highlighted in yellow):

Computational method. All calculations were conducted via the DFT with the generalized Perdew-Burke-Ernzerhof (PBE), and the projector augmented-wave (PAW) pseudopotential planewave method as implemented in the Vienna *ab initio* Simulation Package (VASP) code. For the PAW pseudopotential, $3d^{8.4}s^2$, $3d^{7.4}s^1$, $2s^2 2p^4$ and $1s^1$ were treated as valence electrons for Ni, Fe, O and H atoms, respectively. The bulk β -NiOOH structure was optimized within the local-spin-polarized density approximations (LSDA+U, $U_{\text{eff}}=5.3$ eV for Ni and 5 eV for Fe, respectively). The plane wave cutoff energy of 500 eV was set for all calculations and the energy and force convergence were set to 10^{-5} eV and 0.02 eV \AA^{-1} , respectively. Grimme's DFT-D3(BJ) dispersion correction were used in all calculations. A $12 \times 12 \times 10$ Monkhorst-Pack (MP) k-point grid was used for β -NiOOH unit cells geometry optimization calculations. The layered NiOOH structure was optimized using the same basic settings of the bulk β -NiOOH structure optimization except the k-point sampling ($12 \times 6 \times 1$). The optimized unit cell of layered NiOOH was expanded to 6×3 supercells ($a=17.556$ \AA and $b=17.739$ \AA) containing 72 Ni, 72 H and 144 O atoms with a vacuum thickness of 18 \AA for Fe embedding systems. Based on our experimental results, two different Fe doping concentrations were considered in this study, they are $\text{Ni}_{0.95}\text{Fe}_{0.05}\text{OOH}$ and $\text{Ni}_{0.8}\text{Fe}_{0.2}\text{OOH}$ corresponding to computational models of $\text{Fe}_4\text{Ni}_{68}$ and $\text{Fe}_{14}\text{Ni}_{58}$, respectively. (Fig.S6 a-c) It should be noted that we constrained lattice constants and relaxed all atoms in Fe embedding systems simulations. For supercell case, only γ k-point was considered.

For COM route, the thermodynamic correction of Gibbs free energy was implemented by VASPKIT and the relative energy was calculated by:

$$\Delta G(1)=G(\text{O}^*\text{O})+\frac{G(\text{H}_2)}{2}-G(\text{O}^*\text{OH})-eU$$

$$\Delta G(2)=G(^*\text{OO})-G(\text{O}^*\text{O}) \text{ (Light dominated process)}$$

$$\Delta G(3)=G(\text{HO}^*\text{O}_2)+\frac{G(\text{H}_2)}{2}-G(^*\text{OO})-G(\text{H}_2\text{O})-eU$$

$$\Delta G(4)=G(\text{HO}^*\text{OH})+G(\text{O}_2)+\frac{G(\text{H}_2)}{2}-G(\text{HO}^*\text{O}_2)-G(\text{H}_2\text{O})-eU$$

$$\Delta G(5)=G(\text{O}^*\text{OH})+\frac{G(\text{H}_2)}{2}-G(\text{HO}^*\text{OH})-eU$$

5. How to understand the COM mechanism from a theoretical perspective? The authors described “The

deprotonation of *OH is quite similar to that in the AEM route”, can people use COM mechanism to draw a step diagram like AEM or LOM? For example, first step, *OH → second step, ... → xxx step, O₂.

Response: We are very grateful for the referee’s comment. In the previous work, we have shown that the reaction free energy for the deprotonation step in COM route is nearly the same as that in the AEM route for NR-NiOOH model. (*Nature*, 2022, 611(7937): 702-708.) To further understand the COM mechanism from a theoretical perspective, in this work, the reaction free energies of the Ni_{0.95}Fe_{0.05}OOH and Ni_{0.8}Fe_{0.2}OOH following the COM route, based on the 0V and 1.23 V are calculated. (Fig.R2-4, the calculated method is follow the previous work *Nature*, 2022, 611(7937): 702-708.) The results reveal that both the reaction free energies of deprotonation step and light dominated (O-O) coupling step of Ni_{0.8}Fe_{0.2}OOH with stronger e_g^* band broadening extent are lower than that of Ni_{0.95}Fe_{0.05}OOH, which is consistent with the experimental results. As such, the calculated step diagram following COM route further suggest that a stronger e_g^* band broadening extent could both facilitate *OH deprotonation and photon utilization.

Fig.R2-4. Calculated reaction free energies of the (a) Ni_{0.95}Fe_{0.05}OOH and (b) Ni_{0.8}Fe_{0.2}OOH following the COM route, based on the 0V, 1.23 V. (Fig.S11 in supplementary materials)

To address the comment, we have revised our manuscript as follows

(line 395, page 14, highlighted in yellow): For COM route, the thermodynamic correction of Gibbs free energy was implemented by VASPKIT and the relative energy was calculated by:

$$\Delta G(1) = G(O^*O) + \frac{G(H_2)}{2} - G(O^*OH) - eU$$

$$\Delta G(2) = G(^*OO) - G(O^*O) \quad (\text{Light dominated process})$$

$$\Delta G(3) = G(\text{HO}^*\text{O}_2) + \frac{G(\text{H}_2)}{2} - G(^*\text{OO}) - G(\text{H}_2\text{O}) - eU$$

$$\Delta G(4) = G(\text{HO}^*\text{OH}) + G(\text{O}_2) + \frac{G(\text{H}_2)}{2} - G(\text{HO}^*\text{O}_2) - G(\text{H}_2\text{O}) - eU$$

$$\Delta G(5) = G(\text{O}^*\text{OH}) + \frac{G(\text{H}_2)}{2} - G(\text{HO}^*\text{OH}) - eU$$

(line 214, page 8, highlighted in yellow): Meanwhile, the calculated reaction free energies of the $^*\text{OH}$ deprotonation process for $\text{Ni}_{0.8}\text{Fe}_{0.2}\text{OOH}$ following the COM route is also lower than that of the $\text{Ni}_{0.95}\text{Fe}_{0.05}\text{OOH}$, which is consistent with the pulse-voltammetry results. (Fig.S11)

(line 240, page 8, highlighted in yellow): From a calculated perspective, the reaction free energy of the light dominated step for $\text{Ni}_{0.8}\text{Fe}_{0.2}\text{OOH}$ is 0.89 eV, much lower than that of the $\text{Ni}_{0.95}\text{Fe}_{0.05}\text{OOH}$ (0.94 eV). (Fig.S11)

6. The definition of x in $\text{Ni}_{1-x}\text{Fe}_x\text{OOH}$ is unclear, e.g., in line 67, 85 is " $\text{Ni}_{1-x}\text{Fe}_x\text{OOH}$ (x = 0, 0.05, 0.2) " while line 105 is " $\text{Ni}_x\text{Fe}_{1-x}\text{OOH}$ (0, 0.05, 0.2) ", which is correct?

Response: We are very grateful for the referee's comment. $\text{Ni}_{1-x}\text{Fe}_x\text{OOH}$ (x=0, 0.05, 0.2) was correct and we have corrected all of these in the revised manuscript.

7. Why did you choose x in this range (x = 0, 0.05, 0.2)/ (0, 0.05, 0.2)? Is there a special reason? What happens when x=1?

Response: We are very grateful for the referee's comment. There was no special reason for the selection of 5% and 20%. Here, these two samples, i.e., 5% and 20%, were mainly used to study the effect of Fe dopants induced e_g^* band broadening on the COM. Other ratios, such as x = 40%, 60%, and 100% were also considered. However, the phase of Fe_2O_3 appeared. (Fig.R2-4) Since the Fe_2O_3 was a widely studied photocatalyst, the appearance of Fe_2O_3 might have an influence on the COM results. Hence, in our work, to focus on studying the effect of Fe dopants induced e_g^* band broadening on the COM, we chosen $\text{Ni}_{1-x}\text{Fe}_x(\text{OH})_2$ (x=0, 0.05, 0.2) as model samples.

Fig.R2-4 XRD patterns of hydrothermal derived samples with ratio of Ni : Fe = (1-x) : x, where $x=0.4$, 0.6, 1. (Fig.S1 in supplementary materials)

To address the comment, we have revised our manuscript as follows (line 85, page 3, highlighted in yellow):

It was revealed that 5% and 20% Fe could be successfully doped into the $\text{Ni}(\text{OH})_2$ lattice. More content of Fe dopants *i.e.*, 40%, 60%, 100% would lead to the aggregation of Fe cations to form Fe_2O_3 . (Fig.S1) Since the Fe_2O_3 is one kind of widely studied photocatalyst, the appearance of Fe_2O_3 might affect the COM-based electrochemical results. Hence, to focus on studying the effect of Fe dopants induced e_g^* band broadening on the COM, the $\text{Ni}_{1-x}\text{Fe}_x\text{OOH}$ ($x = 0, 0.05, 0.2$) are chosen as model samples.

8. How to distinguish whether the effect on broadening of the e_g^* band of $\text{Ni}_{1-x}\text{Fe}_x\text{OOH}$ is Fe doping or strain effect?

Response: We are very grateful for the referee's comment. We agree that both doping effect and strain effect can induce NiO_6 distortion, leading to the e_g^* band broadening. (Energy & Environmental Science, 2023, 16(2): 641-652.) The strain effect is often greatly related to the particle size control or the confinement environment modification. In our work, the $\text{Ni}_{1-x}\text{Fe}_x(\text{OH})_2$ ($x=0, 0.05, 0.2$) were prepared via the same hydrothermal parameters (120 °C for 10 h) without using any confinement procedures. Based on FT-EXAFS spectra (Fig.R2-5), neither the peak intensity around 6 Å nor the Ni-Ni bond peak intensity show great difference, which indicates that there is no significant particle size difference among these samples. Hence, the structural difference among these $\text{Ni}_{1-x}\text{Fe}_x(\text{OH})_2$ ($x=0, 0.05, 0.2$) should be ascribed to the doping effect

instead of strain effect. Moreover, in the DFT calculations, it is found that the higher amount of Fe can induce greater NiO₆ distortion, leading to stronger e_g^* band broadening, which is consistent with the Ni *K*-edge XAS and FT-EXAFS results. Hence, for the Ni_{1-x}Fe_x(OH)₂ ($x=0, 0.05, 0.2$) system, the e_g^* band broadening would be ascribed to the different Fe dopants induced various extent of NiO₆ distortion rather than strain effect.

Fig.R2-5 FT-EXAFS spectra of Ni *K* edges of Ni_{1-x}Fe_x(OH)₂ ($x = 0, 0.05, 0.2$), inset showing the enlarged FT results within the range of 1.5 to 1.75 Å. (Fig 3d in revised manuscript)

According to the comment, we have revised our manuscript as follows (line 180, page 6, highlighted in yellow): Neither the peak intensity around 6 Å nor the Ni-Ni bond peak intensity show great difference, which indicates that there is no significant particle size difference among these samples.²² Hence, the structural difference among these Ni_{1-x}Fe_x(OH)₂ ($x=0, 0.05, 0.2$) should be ascribed to the doping effect instead of strain effect.

Reviewer #3

Remarks to the Author: In this work, the authors systematically investigate the coupled oxygen evolution mechanism (COM) in NiFeOOH model catalysts. According to their previous study, the light-triggered COM pathway could promote OER activity significantly by realizing both metal and oxygen redox, provided that a unit cell configuration, e.g. NiO₆ octahedral units in NiOOH structure experience sufficient distortion to have modulated electronic states with both metal and oxygen bands around fermi level. In this study, the authors decipher the relationship between eg* band broadening and OER activity enhancement. Such correlation was investigated at two regions of low and high OER potential, where at low potentials eg* band broadening expedites *OH deprotonation as the RDS, while at high potentials it promotes the non-overlapping region between dz² and a_{1g}* states which result in higher photon utilization efficiency. By conducting systematic electro chemical measurements with Fe-doped NiOOH catalysts showing varying degrees of eg* band broadening and thus NiO₆ distortion, Raman and XAS spectroscopies, and DFT calculations, the facilitated proton transfer at *OH deprotonation RDS at low potential as well as light-triggered electron transfer from M-O bands to dz² orbitals are elucidated. The established relationships between tuned electronic states and improved OER at both low and high potentials are demonstrated for more complex structures to show the universal applicability of COM in OER catalyst development. Overall, this work is novel with significant analyses and discussion, thanks to a suite of advanced measurements and modeling. I would recommend accepting this work after revision of some issues listed below.

1- In page 3, Fig 1a is described as “the highest OER activity is observed for Ni_{0.8}Fe_{0.2}OOH under dark condition”, while it is the opposite in the figure (light condition).

Response: We are very grateful for the referee’s comment. The Ni_{0.8}Fe_{0.2}OOH also showed the higher OER activity than Ni_{0.95}Fe_{0.05}OOH under light condition. To enhance the clarity of the figure, we have made modifications to the color. As shown in Fig.R3-1b (Fig.1b in revised manuscript), the overpotential drops for Ni_{0.8}Fe_{0.2}OOH and Ni_{0.95}Fe_{0.05}OOH at 10 mA cm⁻² between dark and light conditions are 22 mV and 11 mV, respectively. The current density improvement normalized to ECSA and loading mass at an overpotential of 300 mV of Ni_{0.8}Fe_{0.2}OOH are also much higher than the Ni_{0.95}Fe_{0.05}OOH.

Fig.R3-1 Electrochemical characterization of Ni_{1-x}Fe_xOOH (x = 0, 0.05, 0.2) for the OER under dark and light condition. a LSV polarization curves of Ni_{1-x}Fe_xOOH (x = 0, 0.05, 0.2) based on a backward scan conducted at a scan rate of 0.1 mV s⁻¹. **b** Comparison of the overpotential drop at 10 mA cm⁻², the current density improvement normalized to ECSA at an overpotential of 300 mV, and the current density improvement normalized to loading mass at an overpotential of 300 mV, respectively for Ni_{1-x}Fe_xOOH (x = 0.05, 0.2). (Fig.1 in revised manuscript)

2- The pulse-voltammetry measurements and the corresponded slopes for the understudy structures are missing. Please provide the measurements and calculated slopes.

Response: We are very grateful for the referee's comment. For the pulse-voltammetry measurement, one small piece of electrocatalysts on carbon cloth (0.2 cm²) after electrochemical oxidation was used as the working electrode. The potential was set firstly at a low potential (E_l = 1.40 V) for 80 s to stabilize the electrode surface. Then the potential was turned to a higher potential (E_h=1.42 V) for 12 s and back to E_l for 12 s as one cycle. This cycle was repeated while increasing E_h from 1.42 V to 1.48 V versus RHE in 20 mV/step with constant E_l = 1.40 V. The transferred charge normalized to ECSA during each cycle was evaluated by integrating the current pulse/ECSA over time. The results for pulse-voltammetry measurements are provided in Fig.R3-2 and Fig.R3-3.

Fig.R3-2 Pulse voltammetry measurements for (a) $\text{Ni}_{0.95}\text{Fe}_{0.05}\text{OOH}$ and (b) $\text{Ni}_{0.8}\text{Fe}_{0.2}\text{OOH}$ (Fig.S4 in revised manuscript)

Fig.R3-3 Pulse-voltammetry measurements and the integrated charge versus potential for (a) (b) NiS₂-NiOOH, (c) (d) NiSe₂-NiOOH, and (e) (f) Ni₅P₄-NiOOH, respectively. (Fig.S13 in supplementary material)

To address the comment, we have revised our manuscript as follows:

(line 354, page 13, highlighted in yellow): **Pulse-Voltammetry measurement.** One small piece of electrocatalysts on carbon cloth (0.2 cm²) after electrochemical oxidation was used for the pulse-

voltammetry measurement. The potential was set firstly at a low potential ($E_l = 1.40$ V) for 80 s to stabilize the electrode surface. Then the potential was turned to a higher potential ($E_h = 1.42$ V) for 12 s and back to E_l for 12 s as one cycle. This cycle was repeated while increasing E_h from 1.42 V to 1.48 V versus RHE in 20 mV/step with constant $E_l = 1.40$ V. The transferred charge normalized to ECSA during each cycle was evaluated by integrating the current pulse/ECSA over time.

(line 126, page 4, highlighted in yellow): Therefore, pulse-voltammetry measurements are conducted to investigate the *OH deprotonation ability of $Ni_{1-x}Fe_xOOH$ ($x = 0.05, 0.2$) at low potential region. (Fig.S4, detailed protocol is described in the Experimental section).

(line 274, page 10, highlighted in yellow): At the same time, both the *OH deprotonation ability and photon utilization efficiency of X-NiOOH (X= NiS_2 , $NiSe_2$, Ni_5P_4) follow the same order: $NiS_2-NiOOH > NiSe_2-NiOOH > Ni_5P_4-NiOOH$ (Fig.5c and d, Fig.S13).

3- Could the authors discuss over the reported findings and the concept of NiO_6 distortion, *OH deprotonation, and e_g^* band broadening in normal ambient electrolysis conditions? If no light/dark conditions are applied, would the found relationships happen at all and thus impact AEM or even LOM pathways for promoting OER?

Response: We are very grateful for the referee's comment. The concept of NiO_6 distortion, *OH deprotonation and e_g^* band broadening under dark conditions (AEM pathway) were discussed in our previous work (*Energy & Environmental Science*, 2023, 16(2): 641-652.). We prepared three typical Ni-based pre-catalysts, *i.e.*, NiS_2 , $NiSe_2$, and Ni_5P_4 , and then fully converted them into NiOOH by performing pro-longed chronopotentiometry treatment. The obtained NiOOH showed different intrinsic OER activities in the order of $NiS_2-NiOOH > NiSe_2-NiOOH > Ni_5P_4-NiOOH > NiOOH$. (Fig.R3-4) Then, detailed X-ray absorption structure analyses unveiled different extents of strain in X-Ni(OH)₂ (X= NiS_2 , $NiSe_2$, Ni_5P_4), in the order of $NiS_2-Ni(OH)_2 > NiSe_2-Ni(OH)_2 > Ni_5P_4-Ni(OH)_2 > Ni(OH)_2$, which was greatly related to particle size. Ni *K*-edge combined with O *K*-edge XAFS showed that **a stronger strain would lead to a greater extent of NiO_6 octahedron distortion in X-Ni(OH)₂, which resulted in a greater extent of e_g^* band broadening.** (Fig.R3-5) **The stronger e_g^* band broadening would significantly facilitate the electron transfer ability from electrocatalysts to external circuit, leading to higher OER activities (AEM-pathway).** (Fig.R3-6)

Moreover, this relationship might also be applied to the LOM case. As for the LOM process, the deprotonation step is usually the rate-determining step. According to our previous work, when the eg^* band becomes broadening, it would favor the electron transfer from the electrocatalysts to external circuit through deprotonation step. Theoretically, the concept of NiO_6 distortion, *OH deprotonation, and eg^* band broadening could be also applied to the LOM process.

Fig.R3-4 OER activities of the reconstruction-derived NiOOH. (a) OER polarization curves of X–NiOOH (X = NiS₂, NiSe₂, Ni₅P₄) based on a backward scan conducted at a scan rate of 0.1 mV s⁻¹ with NiOOH (obtained from the electro-oxidation of hydrothermally prepared Ni(OH)₂) as the benchmark. (b) Tafel plots derived from OER polarization curves. (c) OER polarization curves of X–NiOOH (X = NiS₂, NiSe₂, Ni₅P₄) normalized to ECSA. (d) OER polarization curves of X–NiOOH (X = NiS₂, NiSe₂, Ni₅P₄) normalized to mass loading. The loading masses of NiS₂–NiOOH, NiSe₂–NiOOH, Ni₅P₄–NiOOH and NiOOH were 1.06 mg cm⁻², 1.10 mg cm⁻², 0.97 mg cm⁻², and 0.95 mg cm⁻², respectively. Reproduced with permission. Copyright 2023, The Royal Society of Chemistry. (*Energy & Environmental Science*, 2023, 16(2): 641-652.)

Fig.R3-5 Structural characterization of X-Ni(OH)₂ (X = NiS₂, NiSe₂, Ni₅P₄). **a.** Normalized Ni K-edge XANES spectra of X-Ni(OH)₂ (X= NiS₂, NiSe₂, Ni₅P₄) with Ni(OH)₂ as the benchmark (inset showing the enlarged results within the range of 8344 to 8352 eV). **b.** White line intensity (obtained from Fig.3a) comparison for hydrothermal prepared Ni(OH)₂. Lower intensity suggests stronger NiO₆ distortion. **c.** FT-EXAFS spectra of Ni K-edges of X-Ni(OH)₂ (X= NiS₂, NiSe₂, Ni₅P₄) and Ni(OH)₂ (inset showing the enlarged FT results within the range of 1.4 to 1.85 Å). **d.** Normalized O K-edge XAS spectra of X-Ni(OH)₂ (X= NiS₂, NiSe₂, Ni₅P₄) and hydrothermal prepared Ni(OH)₂. **e.** Schematic diagram of the effect of NiO₆ octahedron distortion on e_g* band broadening for X-Ni(OH)₂ (X= NiS₂, NiSe₂, Ni₅P₄) with standard Ni(OH)₂ (no distortion) as the benchmark. Reproduced with permission. Copyright 2023, The Royal Society of Chemistry. (*Energy & Environmental Science*, 2023, 16(2): 641-652.)

Fig.R3-6 The effects of e_g^* band broadening in reconstruction-derived NiOOH on OER activity. (a) A schematic illustration of the broadening of the empty upper- and filled lower-Hubbard bands (denoted as UHB and LHB, respectively) induced by NiO_6 octahedron distortion. (b) A schematic illustration of e_g^* band broadening on facilitating the electron transfer during the deprotonation step. (c) The correlation between the NiO_6 octahedron distortion extent and the corresponding electron transfer ability for X–NiOOH (X = NiS_2 , NiSe_2 and Ni_5P_4). (d) The correlation between the electron transfer ability and corresponding OER activities for X–NiOOH (X = NiS_2 , NiSe_2 and Ni_5P_4). Reproduced with permission. Copyright 2023, The Royal Society of Chemistry. (*Energy & Environmental Science*, 2023, 16(2): 641–652.)

To address the comment, we have revised our manuscript as follows (line 300, page 12, highlighted in yellow): Previously, it was also unveiled that the stronger e_g^* band broadening would trigger the emergence of more electronic states around the Fermi level. This effect notably facilitated the electron transfer from electrocatalysts to external circuit via $^*\text{OH}$ deprotonation, resulting in higher OER activities (AEM-pathway).²² As for the LOM route, the deprotonation step is usually the RDS. The conceptual framework encompassing NiO_6 distortion, e_g^* band broadening, and $^*\text{OH}$ deprotonation also holds the potential for application in the context of the LOM process, thereby enabling the realization of enhanced catalytic performance. Therefore, it is believed that increasing the degree of e_g^* band broadening could be a crucial factor on promoting the participation of COM under light irradiation and even provide valuable guidelines for future design of highly efficient OER catalysts following various reaction pathways.

4- The authors should elucidate and discuss why necessarily the established relationships between e_g^* band broadening, *OH deprotonation, and photon utilization promote OER for sulfur compared to selenium and phosphorous. Why having NiS₂ interfaced with NiOOH would instigate such correlations?

Response: We are very grateful for the referee's comment. There are two factors determining e_g^* band broadening, *i.e.*, cation dopants and strain effect. (*Energy & Environmental Science*, 2023, 16(2): 641-652.) In the Fe doped NiOOH system, it is revealed that the Fe dopants could effectively induce NiO₆ octahedron distortion, leading to e_g^* band broadening. The e_g^* band broadening could facilitate the *OH deprotonation and photon utilization, hence promoting the COM-based OER activities. To examine the universality of this concept, it is necessary to verify that in the strain effect system. Hence, the reconstruction derived X-NiOOH (X=NiS₂, NiSe₂, and Ni₅P₄) system is employed, which has different extent of e_g^* band broadening due to the strain effect.

To elucidate the reasons for establishing relationships between e_g^* band broadening, *OH deprotonation, and photon utilization promote OER for X-NiOOH (X=NiS₂, NiSe₂, and Ni₅P₄), we have revised our manuscript as follows (line 260, page 10, highlighted in yellow): Our previous work revealed that the e_g^* band broadening could be induced by both cation dopants and strain effect.²² Hence, to validate the proposed hypothesis, it is necessary to verify that in the strain effect system. Specifically, the reconstruction derived X-NiOOH X=NiS₂, NiSe₂, and Ni₅P₄ samples are prepared through the electro-oxidation of NiS₂/NiSe₂/Ni₅P₄ at a current density of 10 mA cm⁻² for 10 hours, which exhibited different extent of e_g^* band broadening due to the strain effect, with the following order: NiS₂-NiOOH>NiSe₂-NiOOH >Ni₅P₄-NiOOH >NiOOH.²² Here, the OER activities of X-NiOOH (X=NiS₂, NiSe₂, and Ni₅P₄) are investigated under both light and dark conditions.

5- The language should be improved.

Response: We are very grateful for the referee's comment. According to the reviewer's advice, we have improved the language in this revised manuscript.

REVIEWERS' COMMENTS

Reviewer #1 (Remarks to the Author):

The authors well addressed the comments. I have no further questions.

Reviewer #2 (Remarks to the Author):

The authors have answered all my concerns carefully. I would like to recommend it to be published as this version.